# Intrinsic Dimension Correlation: Uncovering Nonlinear Connections in Multimodal Representations

**Lorenzo Basile** [1,2]     **Santiago Acevedo** [3]     **Luca Bortolussi** [1]
**Fabio Anselmi** [1,4]     **Alex Rodriguez** [1,5]

[1]University of Trieste     [2]AREA Science Park     [3]SISSA     [4]MIT     [5]ICTP
lorenzo.basile@phd.units.it

## Abstract

To gain insight into the mechanisms behind machine learning methods, it is crucial to establish connections among the features describing data points. However, these correlations often exhibit a high-dimensional and strongly nonlinear nature, which makes them challenging to detect using standard methods. This paper exploits the entanglement between intrinsic dimensionality and correlation to propose a metric that quantifies the (potentially nonlinear) correlation between high-dimensional manifolds. We first validate our method on synthetic data in controlled environments, showcasing its advantages and drawbacks compared to existing techniques. Subsequently, we extend our analysis to large-scale applications in neural network representations. Specifically, we focus on latent representations of multimodal data, uncovering clear correlations between paired visual and textual embeddings, whereas existing methods struggle significantly in detecting similarity. Our results indicate the presence of highly nonlinear correlation patterns between latent manifolds.

## 1 Introduction

Modern machine learning models have the remarkable ability to extract subtle patterns from complex datasets and use them to perform a wide variety of tasks in an astonishingly accurate way. However, to date, we still lack a complete and accurate understanding of their inner workings, especially in the case of deep neural networks. An active field of research in the interpretability of neural networks is focused on characterizing and quantifying the similarity between different models. To this aim, many works (Raghu et al., 2017; Kornblith et al., 2019; Nguyen et al., 2021) evaluate the statistical correlation between the latent representations produced by the models. This quantification is key because it allows, for example, to disentangle or tie together different aspects of data representations, allowing a better interpretation of how the model makes its decisions. Moreover, assessing the similarity between representations of different models is particularly useful to determine whether the latent spaces are *compatible*, meaning that information extracted by one model can be successfully transferred to others.

This point is crucial particularly when data points are represented by multiple interrelated modalities, such as visual and textual. In recent times, it has been shown (Norelli et al., 2023; Moayeri et al., 2023) that it is possible to build multimodal vision-language models starting from pre-trained unimodal encoders, in an effort to match the outstanding performance of state-of-the-art vision-language models such as CLIP (Radford et al., 2021). These findings indicate that modern deep models can produce compatible representations when evaluated on aligned text-image data, hinting at a strong functional similarity between those. However, we find that standard latent similarity metrics such as CKA (Kornblith et al., 2019) and Distance Correlation (Székely et al., 2007; Zhen et al., 2022) find a very low structural correlation between paired multimodal representations. We hypothesize that this is due to the strongly nonlinear nature of these correlations, making them difficult to analyze with standard methods.

Our work tackles this challenge by introducing a novel metric, dubbed Intrinsic Dimension Correlation ($I_d$Cor) that leverages the concept of intrinsic dimension ($I_d$), i.e., the minimum number of

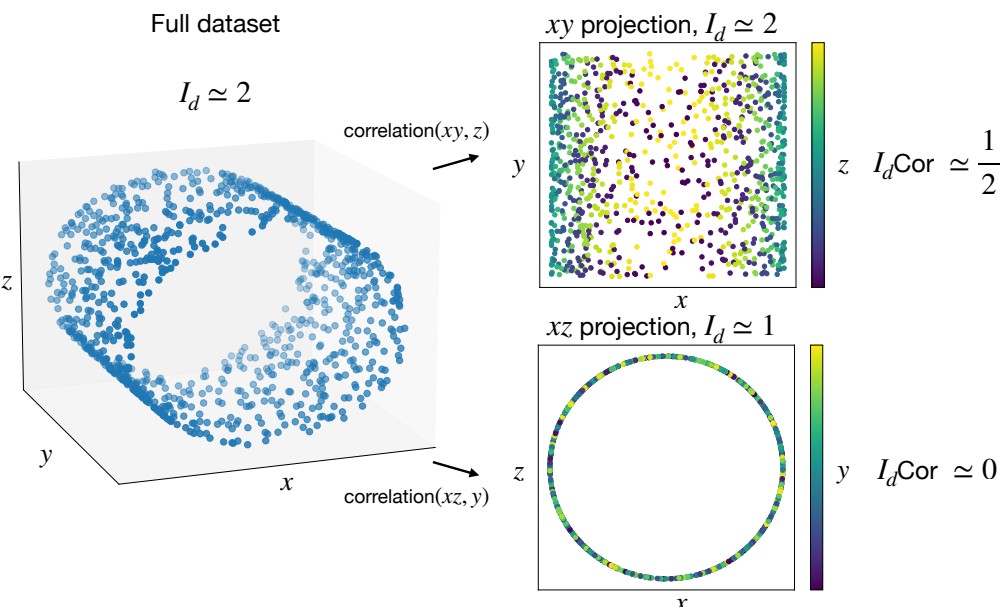

Figure 1: Example usage of $I_d$Cor: we consider a 3D dataset in which the points lie on the surface of a cylinder, hence whose intrinsic dimension ($I_d$) is 2. We want to assess the correlation between the 2D set of coordinates $xy$ (which also has $I_d = 2$, as shown in the top-right panel) and the 1D set $z$. Intuitively, as $x$ and $z$ describe a circle, it is evident that knowledge of $z$ (encoded by color) is very informative in determining the $x$ coordinate (e.g., a yellow point is sure to be found in the central region of the $x$ axis), but not in determining the $y$: hence, the correlation coefficient is $0.5$, according to equation 3. Conversely, when estimating the correlation between $xz$ (whose $I_d$ is 1, bottom-right panel) and $y$, we can see that having access to $y$ (which is now represented by color) does not give any information on the value of $x$ nor $z$, hence the correlation is 0.

variables required to describe the data, to quantify the mutual information between high-dimensional data manifolds. Intuitively, the metric is based on the concept that if two data representations are correlated, the intrinsic dimension of a dataset created by concatenating the features of these representations is reduced, because the information in one representation can describe some aspects of the other. An example is provided in Fig. 1. It is worth noting that, to compute the correlation between two representations, a simpler approach based on the difference between the embedding dimension and the intrinsic dimension of the concatenated dataset would not suffice, since this would measure the amount of correlated features, regardless of whether these correlations are *intra* or *inter* representation. Computational experiments indicate that our method is effective at detecting nonlinear relationships where traditional methods often fail.

Thus, the main contributions of this work can be summarized as follows:

- We propose a novel perspective that links the concepts of correlation and intrinsic dimension. The key idea is that, since the $I_d$ serves as a proxy for the information content of a dataset, the $I_d$ of a dataset created by merging two datasets will represent their joint information.

- Building on this idea, we propose $I_d$Cor, a novel correlation metric based on intrinsic dimension estimation, able to unveil nonlinear correlation between high-dimensional data manifolds, even of unpaired dimensions. We evaluate our metric on synthetic data, showcasing its strengths and limitations in comparison with existing methods.

- We then consider more complex scenarios and quantify the correlation between the representations learned by different deep neural networks engaged in large datasets. Specifically, we focus on multimodal data, demonstrating that we can find strong evidence of correlation where standard methods struggle to identify any.

## 2  BACKGROUND

### 2.1  CORRELATION IN LATENT REPRESENTATIONS

Understanding whether different neural networks can learn to process data in *similar* ways is a crucial point when trying to make sense of their results. Despite the inherent difficulty in defining what it means for two neural networks to be similar (or, at least, to behave similarly), research in this field has made significant progress in the last few years. A well-established approach to test if two neural networks are similar is that of measuring the statistical correlation between the data representations (or embeddings) learned by them. Recent proposals in this direction include Singular Value Canonical Correlation Analysis (SVCCA) (Raghu et al., 2017), Projection Weighted CCA (PWCCA) (Morcos et al., 2018), Centered Kernel Alignment (CKA) (Kornblith et al., 2019), Distance Correlation (dCor) (Székely et al., 2007; Zhen et al., 2022), Aligned Cosine Similarity (Hamilton et al., 2016) and Representation Topology Divergence (RTD) (Barannikov et al., 2022), among others. For a more complete summary of current approaches to neural network similarity measurement, we defer the reader to Klabunde et al. (2023).

These techniques have been widely employed to gain a deeper understanding of various aspects of the way neural models process information: for instance to quantify how different vision architectures encode spatial information (Raghu et al., 2021; Nguyen et al., 2021) or the relation between learning disentangled features and adversarial robustness (Zhen et al., 2022). Taking a slightly different approach, Davari et al. (2023) provides an in-depth analysis of the sensitivity of CKA to transformations that occur frequently in neural latent spaces, showcasing the importance of gathering results from a broader range of similarity metrics to obtain reliable information.

### 2.2  MULTIMODAL LATENT SPACE ALIGNMENT

An example that highlights the importance of assessing similarity, quantified via correlation measures, between neural representations is further demonstrated by the recent empirical findings related to the so-called *latent communication*. This concept, introduced by Moschella et al. (2023), builds on the idea that it is possible to transfer knowledge between latent spaces, even when they are produced by different models and on different data modalities, provided that some semantic alignment between the data exists (for example, images and their textual descriptions). The feasibility of this knowledge transfer was shown in Moschella et al. (2023) through the introduction of *relative* representations, where each point of the original representation is mapped according to its distance from a set of fixed *anchor* points. Using this alternative representation of data, the authors show that it is possible to *stitch* (Lenc & Vedaldi, 2015) together encoders and decoders coming from different models, with little to no additional training.

Furthermore, numerous recent studies have demonstrated that large state-of-the-art visual and textual encoders can produce transferable representations when evaluated on aligned data (i.e., the same data or data that share some semantics, such as image-caption pairs). Indeed, a simple linear transformation is usually enough to map one latent space into another (Moayeri et al., 2023; Merullo et al., 2023; Maiorca et al., 2023; Lähner & Moeller, 2024), at least in terms of performance on a specific downstream task, e.g., classification. It is worth noting that, to perform the alignment of the data, one assumes prior knowledge about the semantic correlation between the data representations (in order to define the anchor points). Hence, while these findings suggest a remarkable similarity between compatible latent spaces, the problem of detecting these correlations without relying on any downstream evaluation is still an open problem. Indeed, our investigation into the connection between aligned textual and visual embeddings reveals a very weak correlation using existing methods, calling for the development of methods that allow the identification of nonlinear correlations in high-dimensional spaces.

### 2.3  INTRINSIC DIMENSION

The concept of the intrinsic dimension ($I_d$) of a dataset is widely used in data analysis and Machine Learning. Before providing a more formal definition, imagine a dataset where your data points are the cities around the globe described by their 3D Cartesian coordinates. We will say that the *embedding dimension* of this dataset is three. However, anyone familiar with cartography would agree that nearly the same information can be encoded with only two coordinates (latitude and

longitude). Therefore, its $I_d$ would be equal to two. Indeed, one of the definitions of $I_d$ is the minimum number of coordinates needed to represent the data with minimal information loss. A complementary definition is the dimension of the manifold in which the data lie, which in this case would be a sphere.

A possible way of estimating the $I_d$ is to find a meaningful projection (i.e., with minimal information loss) into the lowest dimensional space possible. A classical method for doing that is Principal Component Analysis (Wold et al., 1987), but it has the drawback that the intrinsic dimension it estimates is only correct if the manifold in which the data lie is a hyperplane. Therefore, the development of methods that can estimate the $I_d$ in nonlinear manifolds is an active research field. Typically, these approaches infer the $I_d$ from the properties of distances to the Nearest Neighbors. While a full review of these methods is out of the scope of this work (the interested reader is referred to Campadelli et al. (2015)), it is worth mentioning the Maximum Likelihood estimator (MLE) (Levina & Bickel, 2004), the Dimensionality from Angle and Norm Concentration (DANCo) approach (Ceruti et al., 2014) or the TwoNN (Facco et al., 2017). The last is the one employed in this work since it is particularly fast and behaves well even in the case of datasets with a high non-uniformity on the density of points. A brief description of TwoNN is provided in Appendix A.1.

More recently, several studies have estimated the intrinsic dimension of neural representations, demonstrating that $I_d$ is a valuable tool for understanding the geometry of the latent manifolds produced by deep models. This concept was initially explored in Ansuini et al. (2019), where the authors estimated $I_d$ across different layers of CNNs, gaining insights into the sequential information flow within these models. Later, Valeriani et al. (2023) and Cheng et al. (2023) analyzed the representations of transformer models, across different domains, while Kvinge et al. (2023) studied the internal $I_d$ of generative diffusion models. In a slightly different direction, Brown et al. (2022) unveiled a connection between generalization and the $I_d$ of the hidden representations, while Kaufman & Azencot (2023) studied the relation between $I_d$ and curvature in latent manifolds.

## 3 CORRELATION THROUGH INTRINSIC DIMENSION

The intrinsic dimension of a dataset is closely linked to the correlations among the various features that define the data points. These correlations determine the regions in which the data points can exist, thereby shaping the underlying manifold. Let us consider the simplest example: a two-dimensional dataset. If the two variables are uncorrelated, their linear correlation coefficient ($R^2$) approaches zero while, if one feature is a linear function of the other, $R^2$ becomes equal to one. The two scenarios differ by the $I_d$ of the data manifold: the first case corresponds to a plane ($I_d = 2$), while the second corresponds to a line ($I_d = 1$). If we examine a slightly more complex case, the advantage of exploiting the $I_d$ for correlation becomes evident. Let us consider a spiral-shaped dataset embedded in two dimensions: it has $R^2 \approx 0$ due to the nonlinear nature of the correlation between the two variables, while the behavior of the $I_d$ is identical to the one observed on the linearly correlated dataset, as reported in detail in section 4.1.

A formal framework that connects correlation and intrinsic dimension comes from information theory (Romano et al., 2016). In this field, mutual information is the fundamental metric that quantifies the relationship between simultaneously sampled random variables. Mutual information makes a natural candidate to serve as a measure of correlation between data, and Horibe (1985) defined a correlation coefficient between two random variables $X$ and $Y$ (known as Normalized Mutual Information - NMI) as:

$$\varrho = \frac{I(X,Y)}{\max(H(X), H(Y))} \tag{1}$$

where $I(\cdot, \cdot)$ is the mutual information and $H(\cdot)$ is the entropy.

In high dimensional datasets, computing the mutual information is challenging due to the curse of dimensionality, so we introduce the intrinsic dimension as a proxy. Mutual information can be expressed in terms of entropies as:

$$I(X,Y) = H(X) + H(Y) - H(X,Y) \tag{2}$$

Recent research (Bailey et al., 2022; Ghosh & Motani, 2023) has shown that intrinsic dimension is a metric that can satisfy the most desirable properties of entropy, while being easy and fast to compute

on large-scale data (in Appendix A.2, we provide a test case where this relationship is exact). We exploit this relation to define our correlation coefficient $I_d\text{Cor}$, based on the NMI formulation, but using intrinsic dimension as a replacement for entropy. Given two standardized datasets (of possibly different dimensionality) $X \in \mathbb{R}^{n,d_1}$ and $Y \in \mathbb{R}^{n,d_2}$, we define their $I_d\text{Cor}$ as:

$$I_d\text{Cor}(X, Y) = \frac{(I_d(X) + I_d(Y) - I_d(X \oplus Y))}{\max(I_d(X), I_d(Y))} \tag{3}$$

where $\oplus$ denotes row-wise concatenation. Although various other versions of NMI have been proposed over time (for a complete survey on variations of NMI we defer the reader to Vinh et al. (2009)) we adopt the $\max$ normalization since it allows an interpretation as the fraction of features of the most informative dataset that can be predicted using the less informative one (see Fig. 1 for an intuitive explanation of this concept).

In practical applications, intrinsic dimension is calculated using estimators, which can be prone to errors. To mitigate this, we assign a $p$-value to the observed correlation, employing a permutation test (Davison & Hinkley, 1997) on $I_d(X \oplus Y)$. Specifically, we estimate the $I_d$ of several independent samples of the joint dataset, created by concatenating the two original datasets and randomizing the pairings to disrupt any existing correlations. This process allows us to determine a $p$-value that represents the probability of the hypothesis of the two datasets being uncorrelated as $p = \frac{L+1}{S+1}$, where $L$ is the number of estimates lower than the original joint $I_d$ and $S$ is the total number of permuted samples considered.

## 4 RESULTS

In this section, we begin by assessing our proposed $I_d\text{Cor}$ measure in simple, controlled settings using synthetic data. Then, we move to applications to latent representations generated by different neural networks on various datasets. We begin with a straightforward example that underscores the difficulty faced by standard methods in identifying nonlinear relationships, then progress to more extensive applications involving visual representations and multimodal text-image representations. For some experiments, we do not report the correlation $p$-value returned by our method. In such cases, the $p$-value is always the lowest possible according to the permutation test outlined in section 3 ($\frac{1}{S+1}$, where $S$ is the number of permutations, 100 in most of our experiments).

### 4.1 SYNTHETIC EXPERIMENTS

As a first step, we produce three toy datasets, displayed in the Appendix in Fig. 5. Such datasets are made of 5000 observations of two variables that are either linearly correlated, uncorrelated, or nonlinearly correlated. The first setting is simply obtained by arranging $x$ and $y$ on a straight line, in the second case both random variables are sampled independently from a normal Gaussian distribution, while the last dataset contains data arranged on a spiral curve.

We report our correlation results in Table 1, aggregating results over 10 independent random samplings of the datasets. In the simpler cases (linear and random data) our method agrees with linear correlation and distance correlation, correctly identifying a very strong correlation in linear data and the lack thereof in random data. The spiral dataset constitutes a more tricky testbed: while there is a clear correlation between the two variables, it is a highly nonlinear one, and the linear correlation coefficient is around $0$. Even Distance Correlation, despite being a nonlinear method, fails to find any strong signal of correlation, returning a value very close to $0$. Instead, our method correctly identifies the strong dependency between the two variables, with a mean correlation coefficient of $0.98$, determined with high confidence, as witnessed by the $p$-value consistently equal to $0.01$. We note here that our method relies on an intrinsic dimension estimator (in this paper, we mainly employ TwoNN (Facco et al., 2017), but we also provide results on synthetic data using MLE (Levina & Bickel, 2004) in section A.3.1 in the Appendix), and it inherits substantial properties from it. On the negative side, $I_d$ estimators are not oracles, and they can return values that slightly differ from what one would expect (e.g., $I_d$ lower than 1 in our linear dataset or higher than 2 in the random case), or even totally fail (when the $I_d$ becomes large enough, the estimator is also affected by the curse of dimensionality). Conversely, the choice of employing TwoNN makes our method extremely

efficient (the correlation coefficient can be obtained with just 3 calls to the estimator), allowing it to scale easily to large and high-dimensional datasets.

Table 1: Correlation in two-variable datasets. ($\mathbf{R^2}$): linear correlation coefficient; ($\mathbf{dCor}$): distance correlation coefficient; ($\mathbf{I_d \oplus}$): intrinsic dimension of the concatenated (2D) dataset; ($\mathbf{I_d Cor}$): correlation coefficient ($\varrho$) returned by our method; ($\mathbf{p\text{-}value}$): significance of the correlation detected by our method (100 shuffles). Results are reported as mean $\pm$ std over 10 independent random samplings of data, with the exclusion of $p$-value, which is reported as a range.

| Data | $\mathbf{R^2}$ | dCor | $\mathbf{I_d \oplus}$ | $\mathbf{I_d Cor}$ | p-value |
|------|------|------|------|------|------|
| Linear | $1.00 \pm 0.00$ | $1.00 \pm 0.00$ | $0.99 \pm 0.02$ | $1.00 \pm 0.00$ | $0.01$ |
| Random | $0.00 \pm 0.00$ | $0.00 \pm 0.00$ | $2.02 \pm 0.04$ | $-0.02 \pm 0.04$ | $[0.03, 0.99]$ |
| Spiral | $0.01 \pm 0.00$ | $0.02 \pm 0.00$ | $1.01 \pm 0.02$ | $0.98 \pm 0.02$ | $0.01$ |

Then, we switch to a higher-dimensional setting, in which we test our method on correlating pairs of synthetic datasets of 4 variables. We construct a first random dataset containing 4 independently sampled variables, and then consider 3 correlation scenarios: in the first, we compute its correlation with another likewise randomly sampled dataset, hence we expect no correlation; in the second, we randomly sample 2 variables of the second dataset, while binding the other 2 to the original dataset through trigonometric functions, establishing a partial correlation between the two; finally, in the third scenario we set all variables of the second dataset to be trigonometric functions of the original variables, thus making the datasets completely correlated, although in a strongly nonlinear fashion. We compare our $I_d$Cor against Distance Correlation and Canonical Correlation Analysis (CCA) (Hotelling, 1936), which is the direct analogous of $R^2$ in multivariate correlation, and serves as a linear baseline. As we report in Table 2, while all methods agree in correctly finding no correlation in the first scenario, both CCA and dCor fail in the partially and completely correlated cases, while $I_d$Cor returns substantially higher correlation coefficients, showcasing its robustness to nonlinear correlations. In the Appendix (sections A.3.2 and A.3.3), we report additional results in this setting, obtained applying $I_d$Cor to differently sampled and noisy data.

Table 2: Correlation between high dimensional synthetic datasets in various scenarios. We consider a first dataset with 5000 observations of 4 variables $w$, $x$, $y$ and $z$, all sampled independently from a Gaussian distribution with mean 0 and standard deviation $\pi$, and assess correlations in 3 different scenarios. In scenario **A - absence of correlation** we compute the correlation between such dataset and another dataset built by independently sampling corresponding $w'$, $x'$, $y'$ and $z'$ from the same distribution. In scenario **B - partial correlation**, we randomly sample $y'$ and $z'$, while we set $w' = \cos(w + y)$ and $x' = \sin(x)$. Finally, in scenario **C - complete correlation**, we set $w' = \cos(w + y)$, $x' = \sin(x)$, $y' = \sin(xz)$, $z' = \cos(y)$. Results are reported as mean $\pm$ std over 10 independent random samplings of data, with the exclusion of $p$-value, reported as a range.

| Correlation | CCA | dCor | $\mathbf{I_d \oplus}$ | $\mathbf{I_d Cor}$ | p-value |
|------|------|------|------|------|------|
| (A) Absent | $0.02 \pm 0.01$ | $0.00 \pm 0.00$ | $8.07 \pm 0.17$ | $0.01 \pm 0.05$ | $[0.25, 1.00]$ |
| (B) Partial | $0.02 \pm 0.01$ | $0.00 \pm 0.00$ | $6.41 \pm 0.10$ | $0.35 \pm 0.03$ | $0.01$ |
| (C) Complete | $0.02 \pm 0.00$ | $0.01 \pm 0.00$ | $4.88 \pm 0.06$ | $0.67 \pm 0.02$ | $0.01$ |

## 4.2 A MOTIVATING EXAMPLE ON NEURAL REPRESENTATIONS

Across different layers, neural networks encode information in complex, high-dimensional representations that differ significantly from the simpler datasets discussed earlier in this manuscript. In particular, deep models are structured to learn nonlinear functions of the input data, typically through the use of nonlinear activation functions like ReLU. This suggests that the representations produced by different networks on the same data can be correlated in complex, nonlinear ways. Consequently, methods used to detect such correlations need to be capable of capturing this degree of nonlinearity.

To illustrate this phenomenon, we showcase a simple example: we consider a randomly initialized multilayer perceptron (MLP), made of 15 fully connected layers of 784 neurons, followed by a

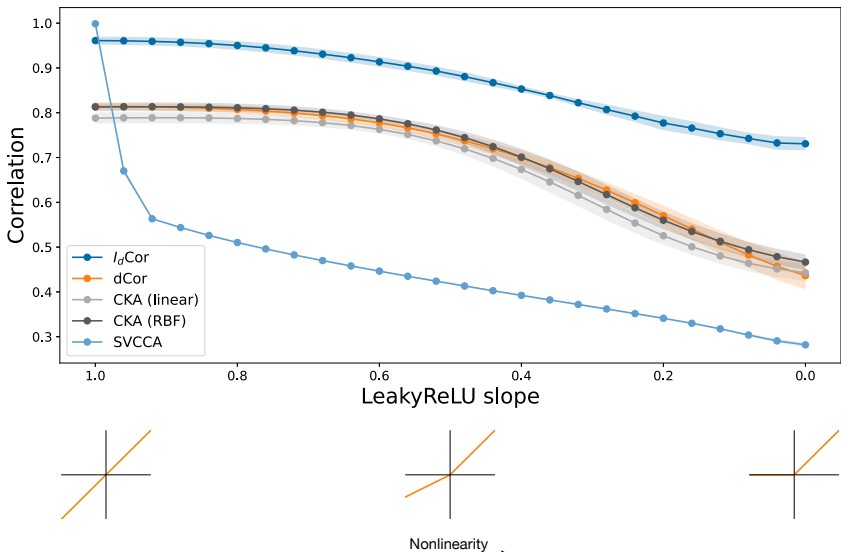

Figure 2: Average correlation results with different methods between MNIST data and their final representations computed by a randomly initialized MLP, with variable degree of activation non-linearity, increasing on the $x$ axis. Shaded area represents standard deviation over 10 runs with independent random initialization of MLP weights.

LeakyReLU activation. LeakyReLU is a parametric activation function, whose behavior depends on a parameter called slope: if the slope is 1, it behaves like the identity function, rendering our MLP a linear function of the input, while lower slope values make the network nonlinear, with 0 corresponding to the standard ReLU. We feed our MLP with the MNIST (LeCun et al., 1998) dataset at increasing degrees of nonlinearity (which corresponds to decreasing the slope) and compute the correlation between the representation at the final layer and the input data, both with our method and with established baselines (SVCCA, Distance Correlation, linear kernel CKA, and RBF kernel CKA).

As we report in Fig. 2, our method is weakly affected by the increasing nonlinearity in the correlation, as it consistently returns correlation coefficients above $0.75$. Existing baselines capture high correlation in linear or quasi-linear cases (high slope), but the signal tends to degrade quickly as the slope decreases. This is especially true for SVCCA, as it is a linear method, but even nonlinear alternatives see the initial correlation fade when the activation becomes ReLU, reaching values below $0.5$.

Our method estimates a proxy of the normalized mutual information between representations, being therefore largely insensitive to nonlinear transformations of the data (as opposed to other metrics like CKA or dCor) and making it optimal to detect statistical dependencies between representations. However, this insensitivity makes it a less informative geometrical similarity index, since the two representations will have a high mutual information regardless of the nature of the nonlinear transformation.

## 4.3 IMAGENET REPRESENTATIONS

Moving to a more realistic setting, we test our method on measuring similarity between ImageNet (Russakovsky et al., 2015) embeddings coming from different neural encoders. We consider a variety of pre-trained architectures, including CNNs (ResNet-18 (He et al., 2016) and EfficientNet-B0 (Tan & Le, 2019)), four variants of Vision Transformers (Dosovitskiy et al., 2020) (including a ViT-CNN hybrid) and three ViT-based self-supervised vision-language models (CLIP-ViT-B (Radford et al., 2021), SigLIP-ViT-B (Zhai et al., 2023) and BLIP-ViT-B (Li et al., 2022)). For all the models, we consider the output of the last hidden representation produced by the encoder, before the classification head: in supervised ViTs this choice corresponds to the last *class* token, while in CNNs

to the pooler output. For CLIP, SigLIP and BLIP, we consider instead the image representation in the shared vision-language space. Since some of the correlation methods we employ as baselines

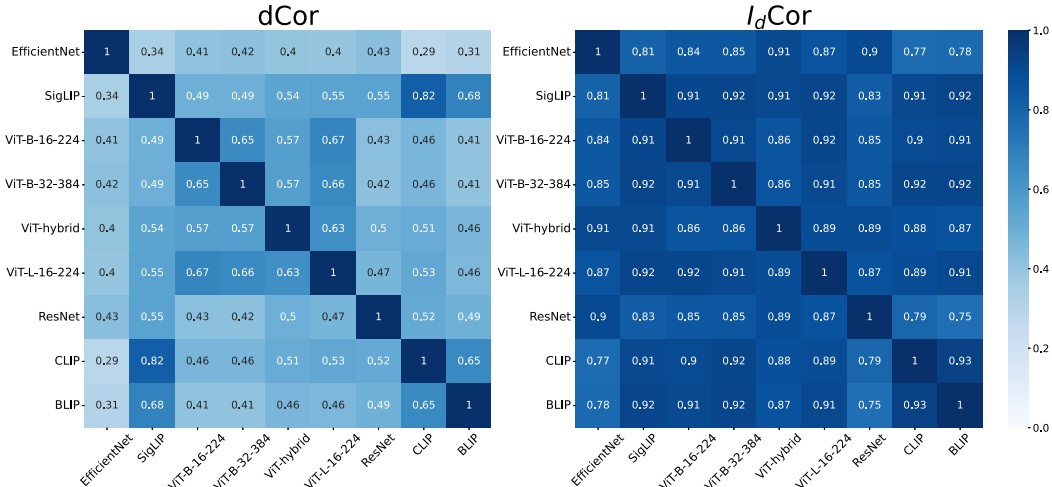

Figure 3: Correlation results on ImageNet representations, obtained using: **left** Distance Correlation (dCor); **right** $I_d$Cor (ours). Both methods are able to detect non-negligible correlation. More baseline results are reported in the Appendix.

(namely, CKA) are particularly expensive in terms of memory requirements, for all our experiments we randomly sample a subset of 30000 data points.

We report the correlation results produced by our method in Fig. 3, along with the correlation scores returned by Distance Correlation (dCor), as it is the baseline method that most closely matches ours in terms of mean off-diagonal correlation ($I_d$Cor mean: $0.87$, dCor mean: $0.50$). Detailed results for SVCCA (mean: $0.46$), linear kernel CKA ($0.43$) and RBF kernel CKA ($0.44$) are reported in the Appendix in Fig. 8. In this setting, all models are computing embeddings for the same data points, hence we would expect significant correlation to be present for any given pair of models. Indeed, all methods are clearly capturing such correlation (even if with higher variance than $I_d$Cor), including SVCCA, which suggests that a relevant component of this correlation is actually linear.

### 4.3.1 COARSE ALIGNMENT

The previous section demonstrated that our method effectively identifies strong correlations within perfectly aligned datasets of representations. We now aim to explore how the performance of the method might vary when applied to coarsely aligned data. To test this, we utilize the inherent class information of ImageNet data. Specifically, we randomly shuffle the embeddings generated by a model while keeping the labels unchanged, and then compare this modified dataset with the original dataset prior to shuffling. In other words, given a point index $i$ of class $C_i$, we pair it with another randomly chosen point $j$ of class $C_j$ with the condition that $C_i = C_j$.

This allows us to assess the robustness of our correlation estimation in less ideal conditions: we expect the correlation signal to decrease, as alignment is a crucial property for all correlation methods to detect similarities. However, as we report in Table 3, we are still able to identify correlations with high confidence, as demonstrated by the low $p$-values. The values of the correlation suggest that the number of features needed to perform the classification task is between 50 and 75%. On average, $I_d$Cor returns a correlation of $0.62$, while baselines sit in the range $0.31 - 0.47$. Interestingly, we observe that the models that exhibit lower correlation are CLIP, BLIP and SigLIP, all contrastive vision-language models. In two of the three cases, $I_d$Cor is even surpassed by Distance Correlation and CKA. While we have no clear explanation for such behavior, these results align with very recent results by Ciernik et al. (2024), highlighting the impact of the pre-training objective on representation similarity. For reference, we also report the $I_d$Cor results obtained when one of the two datasets

is freely shuffled (irrespective of class labels). As witnessed by the high $p$-values, $I_d$Cor correctly reports a lack of correlation in such case.

Table 3: Correlation between ImageNet representations when exact alignment is broken. Results are shown, in terms of $p$-value and correlation coefficient, for the coarse alignment case, in which data are shuffled while preserving the labeling. $I_d$Cor significantly outperforms all baseline methods, with the exception of two models. The last two columns report the $I_d$Cor coefficient and $p$-value for the fully shuffled case, in which no constraint is enforced on labels, and correlation is not present, as identified by high $p$-values. All results are reported as means (excluding $p$-values, reported as ranges) over 5 independent shufflings. Standard deviations are not reported as they are lower than 0.02 in all cases.

| Model | $I_d$Cor | $p$-val. | SVCCA | CKA (lin.) | CKA (RBF) | dCor | $I_d$Cor (rand) | $p$-val. (rand) |
|---|---|---|---|---|---|---|---|---|
| EfficientNet | **0.61** | 0.01 | 0.25 | 0.17 | 0.19 | 0.25 | $-0.10$ | $[0.16, 0.97]$ |
| SigLIP | 0.53 | 0.01 | 0.26 | 0.58 | **0.59** | **0.59** | $-0.01$ | $[0.09, 0.86]$ |
| ViT-B-16 | **0.69** | 0.01 | 0.39 | 0.34 | 0.34 | 0.44 | 0.27 | $[0.27, 0.98]$ |
| ViT-B-32 | **0.68** | 0.01 | 0.40 | 0.40 | 0.40 | 0.47 | 0.22 | $[0.25, 0.99]$ |
| ViT-hyb. | **0.63** | 0.01 | 0.39 | 0.51 | 0.52 | 0.55 | 0.27 | $[0.09, 0.97]$ |
| ViT-L | **0.73** | 0.01 | 0.40 | 0.45 | 0.43 | 0.55 | 0.37 | $[0.18, 0.91]$ |
| ResNet | **0.66** | 0.01 | 0.22 | 0.34 | 0.35 | 0.37 | 0.39 | $[0.46, 0.91]$ |
| CLIP | 0.53 | 0.01 | 0.26 | 0.59 | 0.61 | **0.62** | 0.21 | $[0.08, 0.85]$ |
| BLIP | **0.50** | 0.01 | 0.28 | 0.37 | 0.39 | 0.39 | $-0.10$ | $[0.28, 0.95]$ |
| Average | **0.62** | 0.01 | 0.31 | 0.42 | 0.42 | 0.47 | 0.17 | $[0.08, 0.99]$ |

## 4.4 MULTIMODAL REPRESENTATIONS

We now shift our focus to a multimodal context, where we examine the similarities between hidden spaces learned by text and image encoders. We use three datasets, N24News (Wang et al., 2022), MS-COCO 2014 (Lin et al., 2014) and Flickr30k (Young et al., 2014), all consisting of image-caption pairs. This analysis will help us understand how textual and visual representations correlate when evaluated on related multimodal content. Images are encoded using a representative subset of the vision models introduced in section 4.3: two CNNs (EfficientNet-B0 and ResNet-18), two ViTs (ViT-B-16 and ViT-hybrid), and the visual branch of CLIP. For text we employ five architectures, all taken pre-trained: BERT (Devlin et al., 2019), both cased and uncased, ALBERT (Lan et al., 2019), Electra (Clark et al., 2020) and finally the text encoder of CLIP. For all text models, we consider the last representation of the *class* token.

We report the correlation results obtained on N24News in Fig. 4, comparing our method against Distance Correlation (dCor), which is once again the closest-performing baseline method. Our $I_d$Cor yields a mean off-diagonal correlation of 0.66, which is noticeably higher than those of baseline methods, in the range $(0.25 - 0.29)$. In fact, the correlation heatmaps for all baseline methods reveal a clear block structure, as such methods are able to capture correlation only among same-modality encoders, but fail on cross-modal correlation. Full results for baseline methods are available in the Appendix (Fig. 9). Instead, our method returns significant correlation values even across different modalities, in accordance with previous findings (Maiorca et al., 2023) that proved N24News representations to be transferable across models and modalities. Experiments on Flickr30k and MS-COCO confirm the behavior observed for N24News, as discussed in the Appendix (section A.5.2). Moreover, in sections A.5.3 and A.6, we provide additional experiments on the discussed datasets to show the discriminative power of $I_d$Cor in cases with low or no correlation, while section A.7 contains additional similarity results on N24News, obtained using RTD (Barannikov et al., 2022).

## 4.5 COMPUTATIONAL RESOURCES

We performed all the computations on a NVIDIA A100 GPU, equipped with 40GB of RAM. The main computational hurdle of our method is the estimation of $I_d$ through TwoNN: our implementation follows closely that of Ansuini et al. (2019), which we translated to PyTorch to enable GPU acceleration. With this setup, $I_d$Cor runs in the order of 1s on two 1024-dimensional datasets of 30000 points. Just like our correlation method, all the representation similarity baselines we employ greatly benefit from GPU acceleration: we used them in their PyTorch implementations provided by Miranda (2021) (SVCCA), Maiorca (2024) (CKA) and Zhen et al. (2022) (Distance Correlation), with

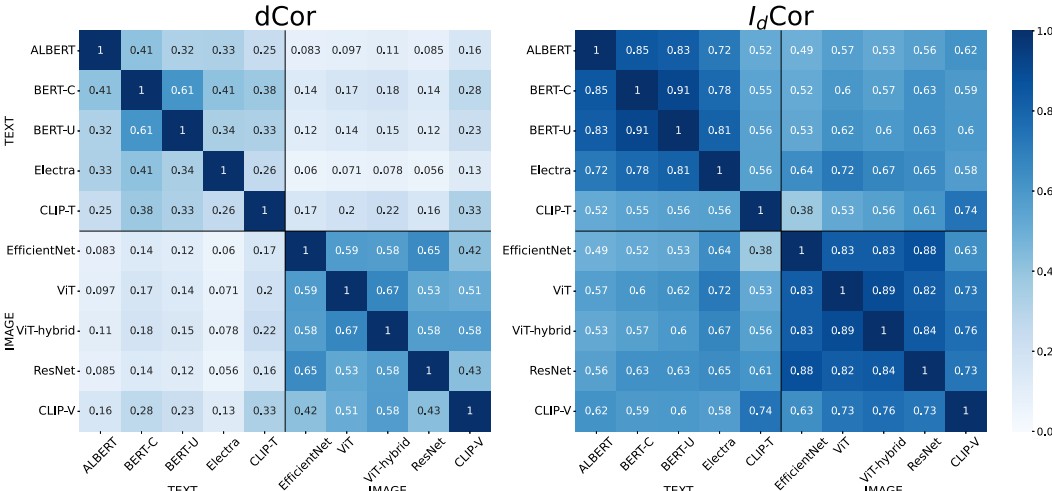

Figure 4: Correlation results on N24News representations, obtained using: **left** Distance Correlation (dCor); **right** $I_d$Cor (ours). Like all other baselines we evaluate, dCor is only able to spot correlations between encoders of the same modality, while $I_d$Cor reveals significant correlation for all model pairs.

minor adaptations. Pretrained models were obtained from the Transformers library by HuggingFace (Wolf et al., 2020), details on the checkpoints we employed are provided in the Appendix (section A.8). Our code is available at https://github.com/lorenzobasile/IDCorrelation.

## 5 DISCUSSION

Our work introduces Intrinsic Dimension Correlation ($I_d$Cor), a novel and robust method for detecting complex nonlinear correlations in high-dimensional spaces. Due to its flexibility, this method can be employed in a wide range of applications, from natural language processing to computer vision and beyond (including other fields of science, like physics), offering a new type of analysis to address how machine learning models represent data. Remarkably, our results show the effectiveness of the method to detect a correlation signal in multimodal data where we know a correlation should exist but where standard methods struggle to identify any.

**Limitations** Among the possible drawbacks of the method, it is worth mentioning that it is fully dependent on the precision of the $I_d$ estimator, so, if the $I_d$ is wrongly predicted the method will fail to find correlations. This would likely happen when the $I_d$s involved are big, so even last-generation estimators will be affected by the curse of dimensionality.

**Future directions** Our method lays the foundations for many interesting future research avenues. For example, it can be used to disentangle data representations by minimizing the correlation between representations from two or more datasets, through the concept of Total Correlation (Watanabe, 1960). This approach could be highly relevant in applications such as multimedia analysis, cross-modal retrieval, and data fusion, potentially resulting in more interpretable neural networks.

In its present form, $I_d$Cor is only applicable to datasets with uniform $I_d$. However, some realistic datasets can present more than one manifold with different intrinsic dimensionalities. With the development of methods that allow estimating in a reliable way these local $I_d$s (Allegra et al., 2020; Dyballa & Zucker, 2023), we devise that $I_d$Cor can be applied to quantify the correlations locally. We provide a proof-of-concept example in the Appendix A.9.

In conclusion, our method not only enhances the understanding of high-dimensional data correlations but also paves the way for innovative solutions in representation learning and interpretability, making it a valuable tool across many fields of machine learning.

ACKNOWLEDGEMENTS

We acknowledge AREA Science Park for making the supercomputing platform ORFEO available for this work. We also thank Sanketh Vedula for insightful discussions on statistical testing. This work is supported by the PNRR project iNEST (Interconnected Nord-Est Innovation Ecosystem) funded by the European Union Next-GenerationEU (Piano Nazionale di Ripresa e Resilienza (PNRR) – Missione 4 Componente 2, Investimento 1.5 – D.D. 1058 23/06/2022, ECS_00000043, CUP J43C22000320006)

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

# A APPENDIX

## A.1 THE TWONN INTRINSIC DIMENSION ESTIMATOR

The TwoNN method (Facco et al., 2017) used in this work is a local intrinsic dimension ($I_d$) estimation technique. Its distinguishing feature is that, by focusing solely on the first two nearest neighbors, it minimizes the size of the $I_d$-dimensional hyperspheres over which the density is assumed to be constant. This method is based on the distribution functions of neighborhood distances, which depend on $I_d$.

Specifically, for each point $x$ in the dataset, it considers the first and second nearest-neighbor distances, denoted as $r_1(x)$ and $r_2(x)$, respectively. Assuming that the dataset is locally uniform within the range of second nearest neighbors, it has been shown by Facco et al. (2017) that the distribution of the ratio $\mu = r_2(x)/r_1(x)$ follows:

$$f(\mu) = I_d \mu^{-I_d - 1}. \tag{4}$$

And using the corresponding cumulative distribution function, $P(\mu)$, one can write:

$$I_d = -\frac{\ln\left[1 - P(\mu)\right]}{\ln\left(\mu\right)}, \tag{5}$$

This relation allows for the estimation of $I_d$ by fitting the set $S = \{(\ln(\mu), -\ln\left[1 - P^{\mathrm{emp}}(\mu)\right]\}$ with a straight line passing through the origin. Here, $P^{\mathrm{emp}}(\mu)$ represents the empirical cumulative distribution, computed by sorting the values of $\mu$ in ascending order.

## A.2 CORRELATION THROUGH THE INTRINSIC DIMENSION: AN EXACT EXAMPLE.

In order to provide an intuition of the relationship between the intrinsic dimension and the entropy of a data set, let us imagine data sets generated in this way: 1) Sample a multivariate Gaussian of dimension $d$ and identity covariance matrix. This can be done by simply generating $d$ series of Gaussian distributed numbers with standard deviation equal to 1. 2) Embed them in a higher dimensional space of dimension $D$ and apply a random angle rotation around a random vector of this dimension.

The entropy of such a dataset could be approximated by the differential entropy of a multivariate Gaussian distribution

$$H = \frac{d}{2}\log(2\pi e) + \frac{1}{2}\log\det(Cov) = \frac{d}{2}\log(2\pi e) \tag{6}$$

where the last equality derives from the use of an identity covariance matrix.

Another interesting property of such a dataset is that, since we have generated the dataset through rotation, the quantity $d$ can be recovered as the number of non-zero eigenvalues of the covariance matrix. Please note that this is equivalent to identifying the intrinsic dimension using PCA.

Now let us imagine that we generate three datasets in this way ($X_1$, $X_2$ and $X_3$) each of them with its own intrinsic dimension $d_1$, $d_2$, $d_3$ and embedding dimension $D_1$, $D_2$, $D_3$. While datasets $X_1$ and $X_2$ are completely independent, dataset $X_3$ is built using one of the series employed for building dataset $X_1$.

We will use PCA for estimating the $I_d$ and apply equation 3 to compute the correlation between datasets these datasets. In the case of datasets $X_1$ and $X_2$, being fully independent, we will have $I_d(X_1) = d_1$, $I_d(X_2) = d_2$, and $I_d(X_1 \oplus X_2) = d_1 + d_2$. By applying PCA, we can recover our datasets as multivariate normal Gaussian, therefore obtaining:

$$\varrho = \frac{H(X_1) + H(X_2) - H(X_1 \oplus X_2)}{\max(H(X_1), H(X_2))} = \frac{\frac{d_1}{2}\log(2\pi e) + \frac{d_2}{2}\log(2\pi e) - \frac{d_1 + d_2}{2}\log(2\pi e)}{\max(\frac{d_1}{2}\log(2\pi e), \frac{d_2}{2}\log(2\pi e))} =$$

$$\frac{d_1 + d_2 - (d_1 + d_2)}{\max(d_1, d_2)} = I_d\mathrm{Cor}(X_1, X_2) = 0 \tag{7}$$

The situation is identical for datasets $X_2$ and $X_3$ but, for $X_1$ and $X_3$, the situation changes a bit. In this case, applying PCA to the combined dataset $X_1 \oplus X_3$ will also provide a multivariate Gaussian, but with a dimension equal to $d_1 + d_3 - 1$, therefore obtaining $\varrho = I_d\text{Cor}(X_1, X_3) = \frac{1}{\max(d_1, d_3)}$.

## A.3 ADDITIONAL RESULTS ON SYNTHETIC DATA

In Fig. 5, we display the two-variable synthetic datasets we used in section 4.1 as a first validation for our method.

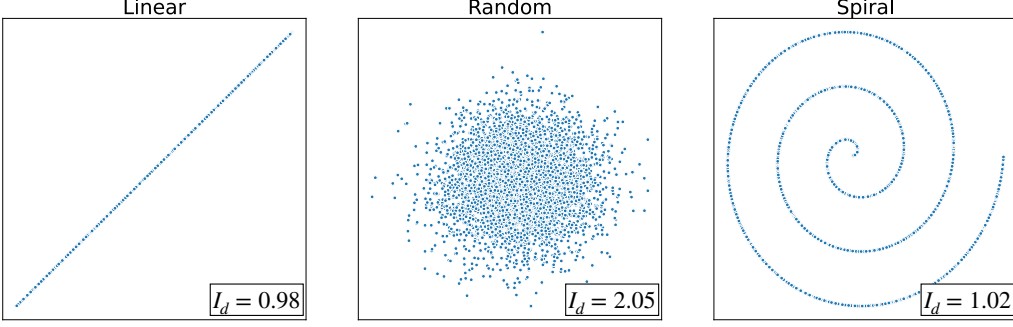

Figure 5: Synthetic datasets, each associated with its intrinsic dimension.

### A.3.1 RESULTS USING THE MLE ESTIMATOR

$I_d\text{Cor}$ is fundamentally powered by an intrinsic dimension estimator. In the main text, we employed TwoNN (Facco et al., 2017), but this is only one of the possible choices. To highlight this point, we provide here results on the synthetic experiments of section 4.1, using the Maximum Likelihood Estimator (MLE) by Levina & Bickel (2004) instead of TwoNN. MLE relies fundamentally on a hyperparameter $k$, the number of nearest neighbors to consider for each data point. As we show in Fig. 6, $k$ can significantly impact the estimated $I_d$ and, in general, there is no clear strategy to choose it. Based on these curves, we opt for $k = 100$ in our experiments, to obtain a reasonable trade-off between efficiency (the lower $k$ the faster MLE is) and accuracy. We report in Table 4 the $I_d\text{Cor}$ results with MLE in the same datasets of Table 1.

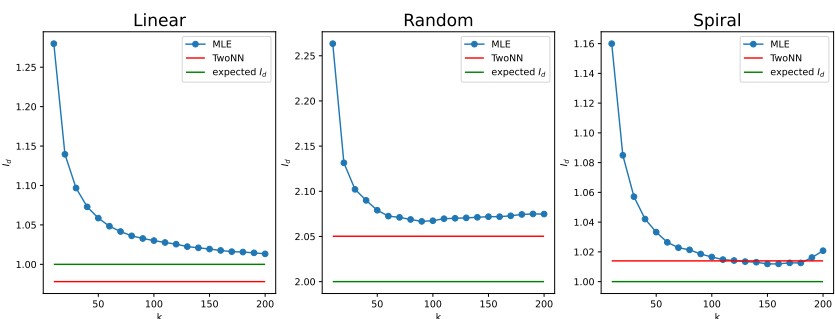

Figure 6: Intrinsic dimension estimation with the MLE estimator, varying the number of neearest neighbors $k$. Horizontal red and green lines report the estimate found by TwoNN and the expected $I_d$ of data, respectively.

In Table 5, we report the results of $I_d\text{Cor}$ equipped with MLE $I_d$ estimation on the same datasets employed in Table 2. We observe that $I_d\text{Cor}$ equipped with MLE produces slightly worse results with respect to TwoNN, as $I_d\oplus$ is estimated less accurately (expected values would be 8, 6 and 4 respectively for scenarios A, B and C).

Table 4: Correlation in the two-variables datasets of Table 1. ($\mathbf{I_d}\oplus$): intrinsic dimension of the concatenated (2D) dataset, computed with MLE; ($\mathbf{I_d Cor}$): correlation coefficient ($\varrho$) returned by our method; (**p-value**): significance of the correlation detected by our method (100 shuffles). Results are reported as mean $\pm$ std over 10 independent random samplings of data, with the exclusion of $p$-value, which is reported as a range.

| Data | $\mathbf{I_d}\oplus$ | $\mathbf{I_d Cor}$ | p-value |
|---|---|---|---|
| Linear | $1.03 \pm 0.00$ | $1.00 \pm 0.00$ | $0.01$ |
| Random | $2.07 \pm 0.00$ | $0.01 \pm 0.00$ | $[0.04, 0.76]$ |
| Spiral | $1.02 \pm 0.00$ | $1.06 \pm 0.00$ | $0.01$ |

Table 5: Correlation between high dimensional synthetic datasets in the scenarios of Table 2, when TwoNN is replaced by MLE in the computation of $I_d$Cor. Results are reported as mean $\pm$ std over 10 independent random samplings of data, with the exclusion of $p$-value, which is reported as a range.

| Correlation | $\mathbf{I_d}\oplus$ | $\mathbf{I_d Cor}$ | p-value |
|---|---|---|---|
| (A) Absent | $7.52 \pm 0.01$ | $0.17 \pm 0.00$ | $[0.17, 0.93]$ |
| (B) Partial | $6.68 \pm 0.01$ | $0.27 \pm 0.00$ | $0.01$ |
| (C) Complete | $5.81 \pm 0.01$ | $0.38 \pm 0.01$ | $0.01$ |

### A.3.2 RESULTS ON A DIFFERENT DATA DISTRIBUTION

In Table 6, we provide the results for the same experiment of Table 2, when data are sample from a uniform distribution in $[-\pi, \pi]$ instead of a Gaussian distribution. From this point, we switch back to the TwoNN estimator.

Table 6: Correlation between high dimensional synthetic datasets in the scenarios of Table 2, when the original dataset is sampled independently and uniformly in $[-\pi, \pi]$. Results are reported as mean $\pm$ std over 10 independent random samplings of data, with the exclusion of $p$-value, which is reported as a range.

| Correlation | CCA | dCor | $\mathbf{I_d}\oplus$ | $\mathbf{I_d Cor}$ | p-value |
|---|---|---|---|---|---|
| (A) Absent | $0.02 \pm 0.01$ | $0.00 \pm 0.00$ | $7.21 \pm 0.12$ | $0.15 \pm 0.04$ | $[0.09, 0.96]$ |
| (B) Partial | $0.17 \pm 0.05$ | $0.06 \pm 0.00$ | $5.74 \pm 0.11$ | $0.48 \pm 0.04$ | $0.01$ |
| (C) Complete | $0.23 \pm 0.02$ | $0.21 \pm 0.00$ | $4.03 \pm 0.06$ | $0.92 \pm 0.02$ | $0.01$ |

### A.3.3 ROBUSTNESS TO NOISE

To evaluate how $I_d$Cor behaves in a more complex and noisy setting, we consider Scenario **C (complete correlation)** from Table 2, and perturb the second dataset $(x', y', w', z')$ with Gaussian noise with mean 0 and increasing standard deviation, up to 3 times larger than the scale of the dataset (all variables are obtained through sinusoidal functions, hence they are constrained in $[-1, 1]$. We report in Fig. 7 the correlation results obtained with $I_d$Cor, averaging the correlation over 5 independent random noise samplings for each standard deviation value. We also report the range of $p$-values for each noise magnitude, showing that correlation is detected with high confidence approximately until the point where noise and signal have similar amplitude.

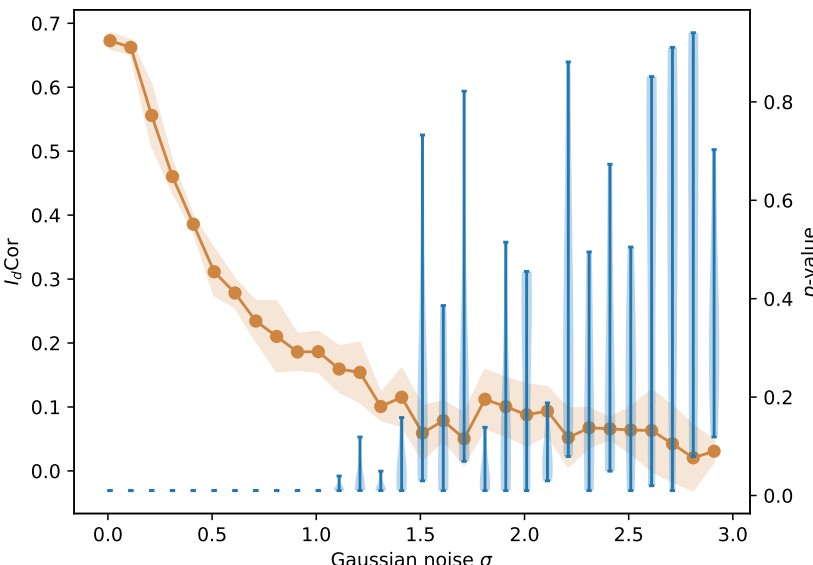

Figure 7: $I_d$Cor results (correlation coefficient and $p$-value) on completely correlated data from Table 2, scenario **C**, perturbed with random Gaussian noise of variable magnitude.

## A.4 ADDITIONAL IMAGENET CORRELATION RESULTS

We provide detailed results for latent correlation in ImageNet (Fig. 8), using SVCCA, linear kernel CKA and RBF kernel CKA.

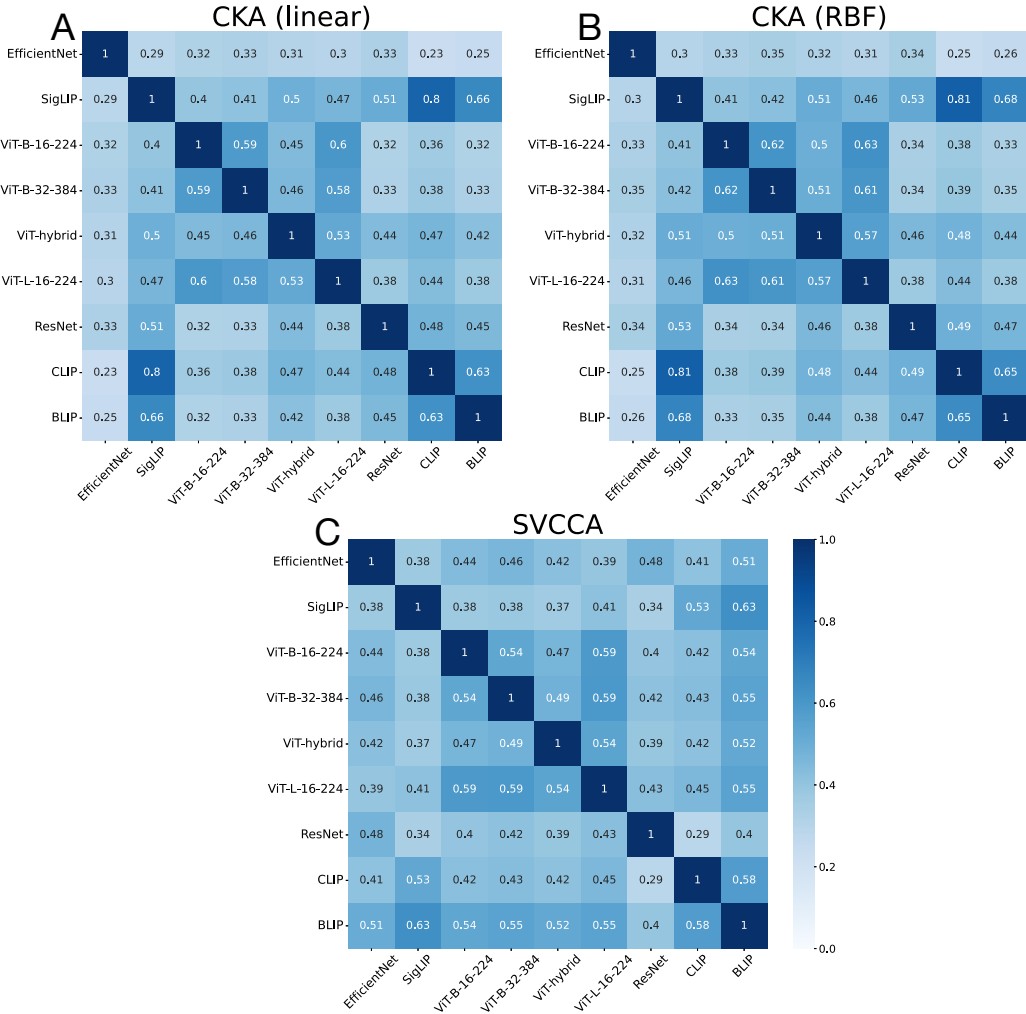

Figure 8: Additional baseline correlation results on ImageNet representations, obtained using: (A) Linear CKA; (B) RBF CKA; (C) SVCCA

## A.5 Additional correlation results on multimodal datasets

### A.5.1 Additional baseline results on N24News

In Fig. 9, we provide additional results for SVCCA, linear kernel CKA and RBF kernel CKA on N24News representations. These methods perform similarly to Distance Correlation, and only capture high correlation between representations belonging to the same data modality.

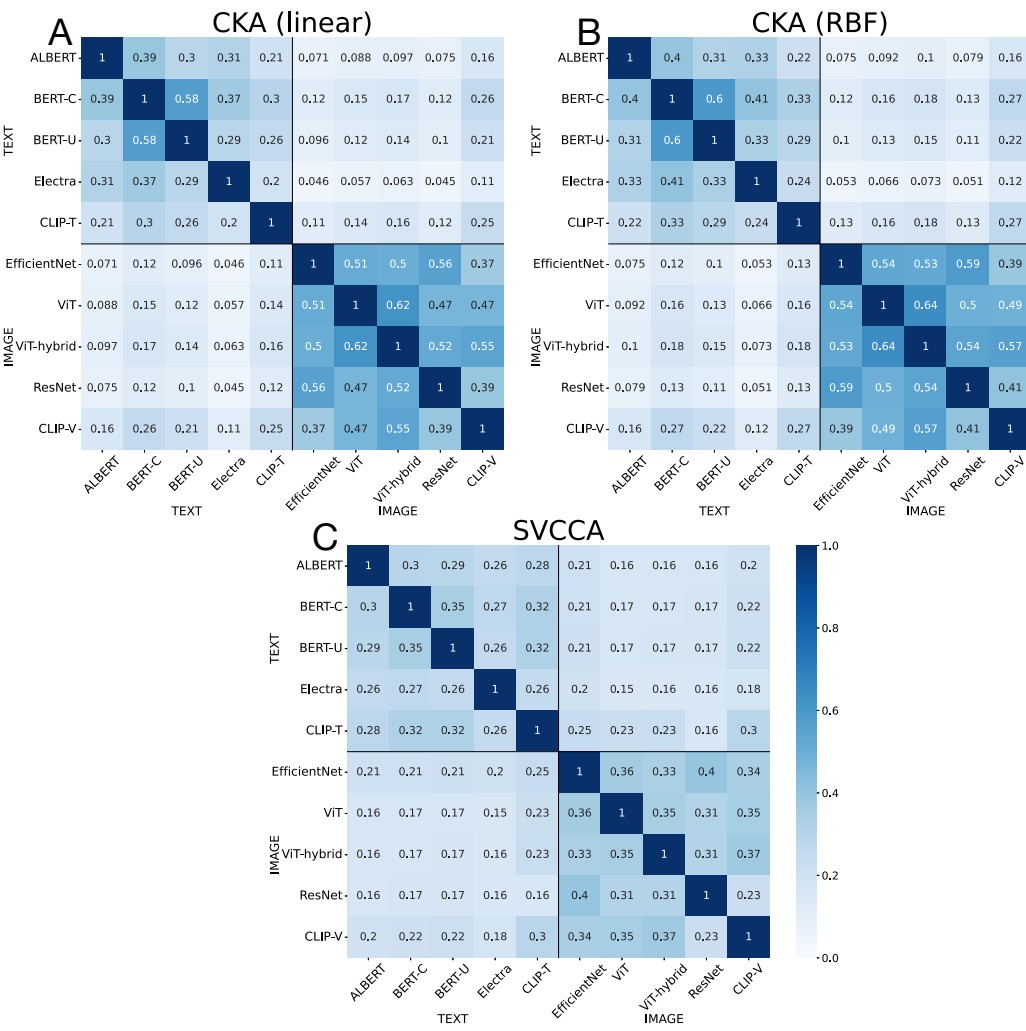

Figure 9: Additional baseline correlation results on N24News representations, obtained using: (A) Linear CKA; (B) RBF CKA; (C) SVCCA

### A.5.2 RESULTS ON OTHER MULTIMODAL DATASETS

In Fig. 10 and Fig. 11 we report the correlation results we obtain respectively on Flickr30k (Young et al., 2014) and MS-COCO (Lin et al., 2014), two multimodal datasets containing images and the corresponding captions. We employ the same models used for N24News (section 4.4). As for the previous dataset, we observe that $I_d$Cor outperforms previous baselines, which significantly struggle to find similarities between embeddings of different modalities.

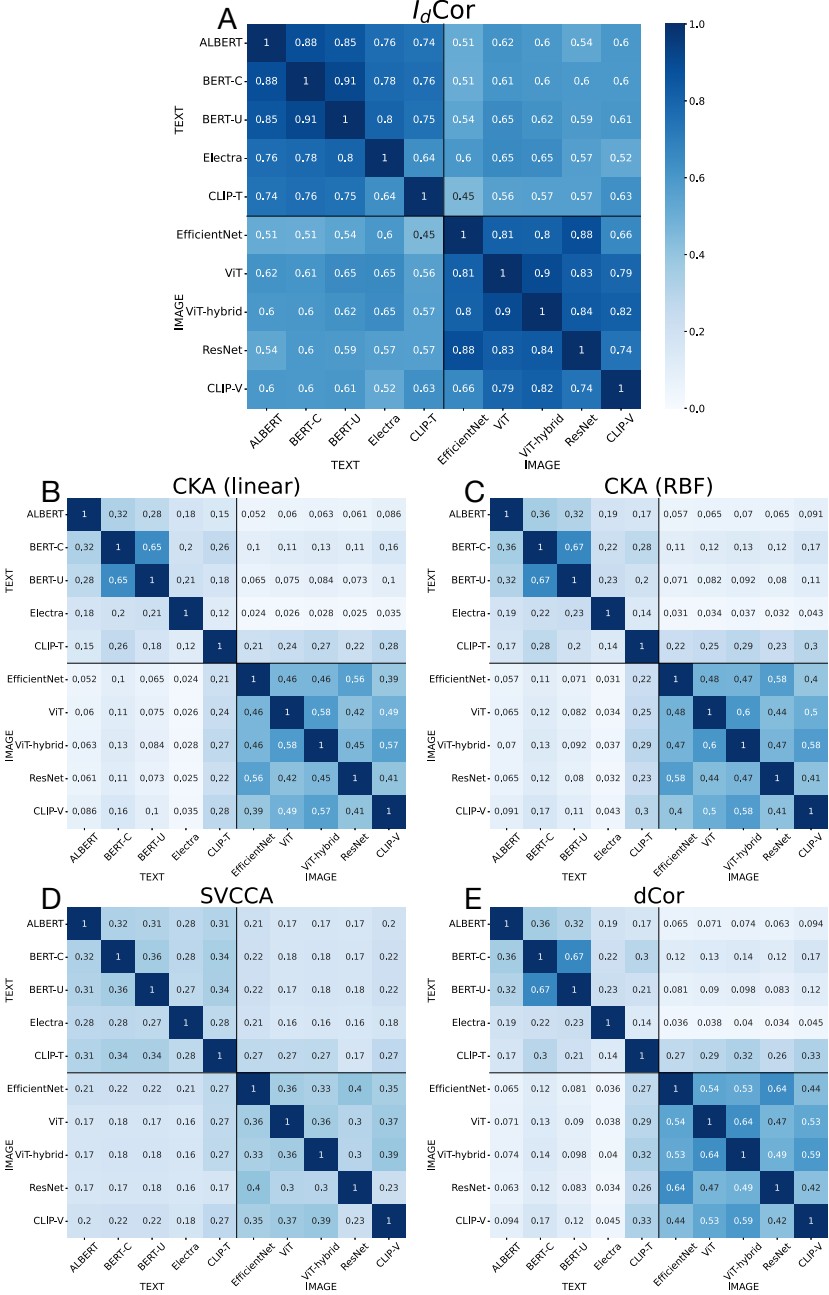

Figure 10: Correlation results on Flickr30k representations, obtained using: (A) our method $I_d$Cor; (B) linear kernel CKA; (C) RBF kernel CKA; (D) SVCCA; (E) Distance Correlation

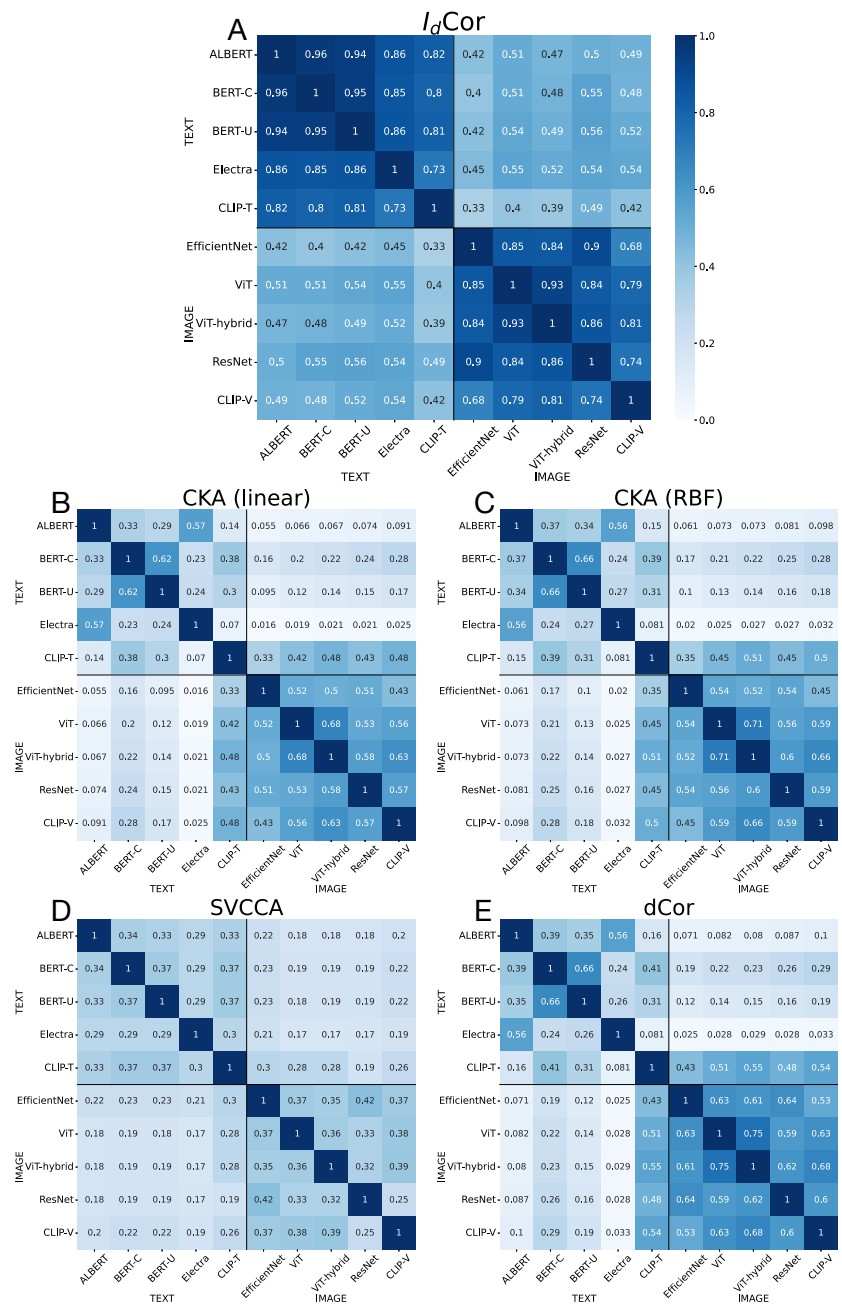

Figure 11: Correlation results on MS-COCO representations, obtained using: (A) our method $I_d$Cor; (B) linear kernel CKA; (C) RBF kernel CKA; (D) SVCCA; (E) Distance Correlation

### A.5.3    PARTIAL MULTIMODAL CORRELATION

In a multimodal setting, $I_d$Cor adapts to the strength of the correlation between textual and visual representation. In this experiment, we consider again the Flickr30k dataset, which provides 5 human annotated captions per image. Instead of computing the correlation between each image representation and a single corresponding caption set, we consider here an enriched textual representation, containing the concatenated encoding for all 5 captions. Then, we progressively perturb the correlation by shuffling an increasing number of textual representations. As we report in Fig. 12, $I_d$Cor decreases almost linearly with the number of perturbed textual representations. Instead, the $p$-value stays at the minimum (0.01) until the last step, where all caption embeddings are shuffled and, consequently, correlation is totally lost.

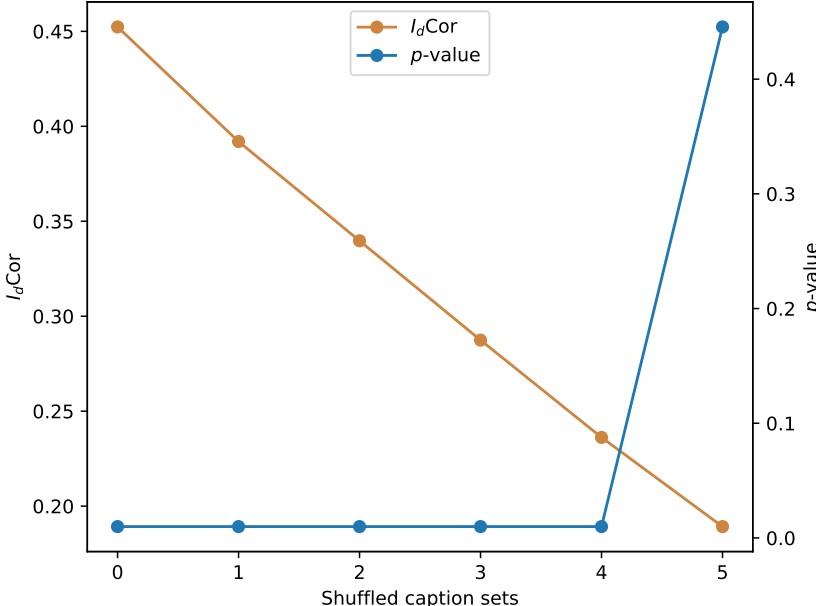

Figure 12: Correlation between visual and textual representations in Flickr30k, considering all 5 captions per image and shuffling an increasing number of textual encoding sets. Representations are computed using CLIP-ViT (images) and BERT (text).

## A.6 CORRELATION BETWEEN DIFFERENT DATASETS

In this section, we provide an assessment of $I_dCor$ on unimodal image representations of different input datasets, computed using CLIP-ViT-B. As we report in Fig. 13, $I_dCor$ scores significantly fall off-diagonal (on different input data), and the corresponding $p$-values increase, indicating lack of correlation.

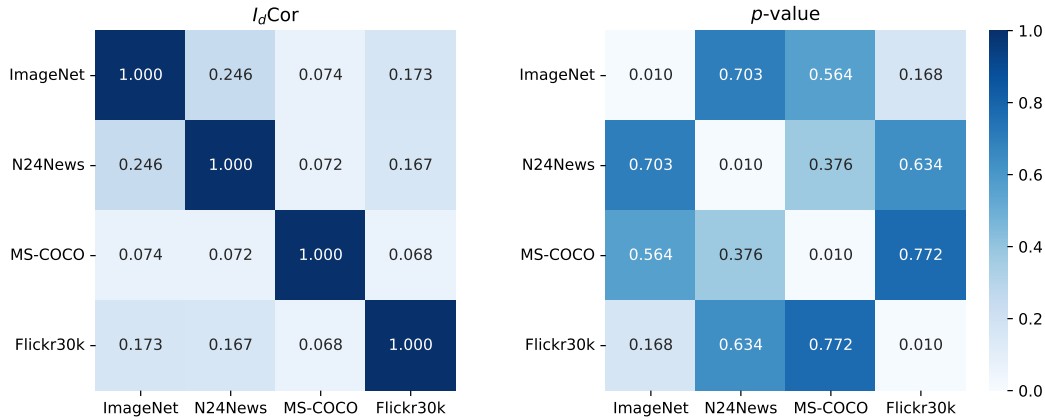

Figure 13: Correlation results between representations produced by the CLIP vision encoder on different image datasets, reported in terms of $I_dCor$ coefficient (left) and $p$-value (right).

## A.7 MULTIMODAL RTD SCORES

In this section, we report the divergence scores computed using Representation Topology Divergence (RTD) (Barannikov et al., 2022) on N24News encodings. As reported in Fig. 14, RTD captures stronger similarities within the visual domain (off-diagonal mean divergence is 26.8), while it finds weaker similarities in the textual and cross-modal settings, with means slightly above 45.

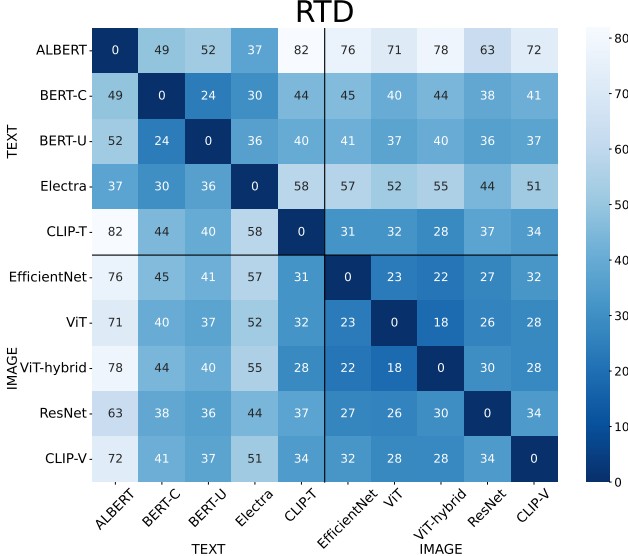

Figure 14: RTD (Barannikov et al., 2022) scores on N24News representations. Colormap is inverted w.r.t. previous figures as RTD produces a divergence score, and not a similarity score.

## A.8 MODEL DETAILS

All models we employ are taken pre-trained from the HuggingFace `transformers` Wolf et al. (2020) library. We report in Table 7 the full list of pre-trained models we employed in this work, associated with the name of the corresponding checkpoint in the library.

Table 7: Reference guide for pre-trained model checkpoints in HuggingFace `transformers` Wolf et al. (2020) library.

| Name in the paper | pre-trained checkpoint name |
|---|---|
| ALBERT | `albert/albert-base-v2` |
| BERT-C | `google-bert/bert-base-cased` |
| BERT-U | `google-bert/bert-base-uncased` |
| Electra | `google/electra-base-discriminator` |
| CLIP (-T/-V) | `openai/clip-vit-base-patch16` |
| BLIP | `Salesforce/blip-itm-base-flickr` |
| EfficientNet | `google/efficientnet-b0` |
| SigLIP | `google/siglip-base-patch16-224` |
| ViT-B-16-224 (ViT) | `google/vit-base-patch16-224` |
| ViT-B-32-384 | `google/vit-base-patch32-384` |
| ViT-hybrid | `google/vit-hybrid-base-bit-384` |
| ViT-L-16-224 | `google/vit-large-patch16-224` |
| ResNet | `microsoft/resnet-18` |

## A.9 LOCAL $I_d$COR

In this section, we provide a proof-of-concept illustration of a local version of $I_d$Cor, that employs IAN (Dyballa & Zucker, 2023), a local $I_d$ estimator.

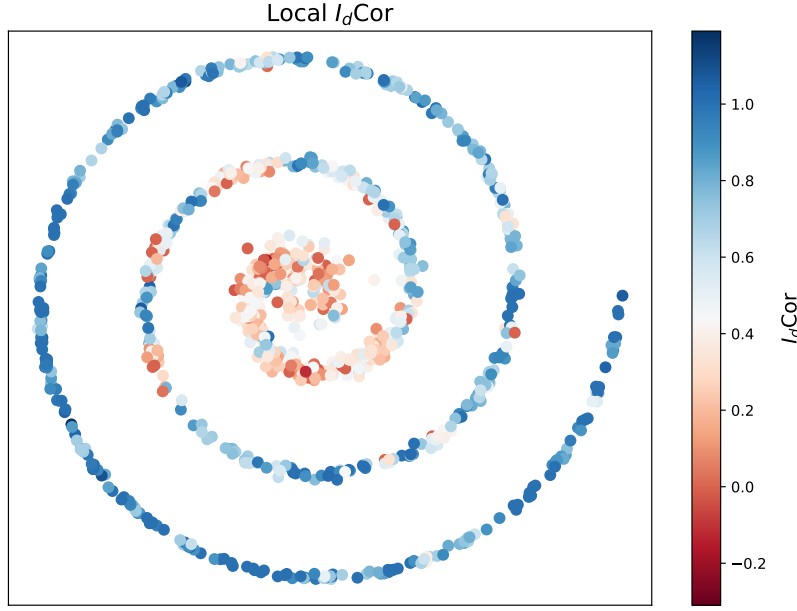

Figure 15: Local $I_d$Cor with IAN estimator. It can be seen that in the center of the spiral, where the data is nearly uniformly distributed, the correlation is low, while in the outer region, where correlations force the data to be in the arms of the spiral, the $I_d$Cor index is 1.

