# OpenReview forum: "Intrinsic Dimension Correlation: uncovering nonlinear connections in multimodal representations"
_ICLR.cc/2025/Conference — ICLR 2025 Poster_

### Official Review · Reviewer_i5Pr · 2024-10-27

**Soundness:** 4
**Presentation:** 4
**Contribution:** 3
**Rating:** 8
**Confidence:** 5

**Summary:**

This paper introduces a novel metric for latent space analysis based on intrinsic dimension correlation. With respect to previous metrics, this paper proposes a method that takes care of data nonlinearities and computations efficiency, two crucial aspects of real-worls problems.

I would like to thank the authors for this interesting paper. Overall, I find the paper interesting and relevant to the scope of the conference.

**Strengths:**

1) The paper explores the latent space from an information-theoretic point of view, which is very valuable considering that nowadays lots of papers just do so from architectural and neural points of view.
2) The method proposed in the paper consider the nonlinearity of the data, representing a crucial advancement with respect to previous methods that mainly focus on linear structures.
3) Results of the method seem consistent across different types of representations, ranging from different models to multimodal.
4) The paper is readable and easily understandable.

**Weaknesses:**

1) Although the authors claim that the proposed method can be extended to more than two modalities, no mathematical formulation is presented for this in the method section. Indeed, equations (1,2,3) focus on X and Y, and I am curious if it is possible to extend this to three or more.
2) A graphical representation, which is lacking, of the differences between the various methods (like figure 1) may help the reader better understand that method and its advantages.
3) The authors could maybe insert a downstream example in which their method makes the difference. This would make the paper more effective and also highlight the importance of the proposal to practical researchers.

Minor concerns that do not affect my rating but that the authors could improve:
1) In scientific writing, the Saxon genitive should be avoided, even though I know that both some LLMs and Grammarly suggest using it. I would personally remove the usage of the Saxon genitive in the paper.
2) Usually, figures of results are presented putting on the left the old method and on the right the proposed one. In this paper, the authors often do the opposite, which may confuse the reader. I suggest changing to the conventional method of results presentation.

**Questions:**

1) Related to the weakness 1, I would like to know whether exists a formulation for more than two modalities.

---

> ### Author Response · Authors · 2024-11-19
>
> We thank the Reviewer for carefully evaluating and appreciating our work. We try here to address the weaknesses that emerged in the review.
>
> **W1 and Q1:** In its current form, our method can only be applied to pairs of modalities. However, we are developing an extension that accommodates more than two modalities by leveraging the concept of Total Correlation [a], a multivariable generalization of Mutual Information, where entire modalities are considered instead of individual coordinates. However, this extension is still a work in progress and not yet ready for publication. If the Reviewer deems it appropriate, we are willing to remove references to this possibility in the final version of our paper.
>
> **W2:** We agree that it would be helpful to have an intuitive graphical comparison between different correlation methods. However, we could not find a suitable graphical representation of the methods we use as baselines, as these offer a less immediate geometrical interpretation than intrinsic dimensionality.
>
> **W3:** We are currently evaluating an application of $I_d$Cor for Cross-Modal Knowledge Distillation [b], in which knowledge is transferred across modalities from a large pre-trained teacher to a smaller student model. $I_d$Cor is a differentiable measure (as long as the $I_d$ estimator is differentiable, see [c] for a recent proposal in this field), and it can be employed as part of the training loss to enforce consistency between the two models. In particular, this setup could be useful in settings with limited data in one of the modalities, where training from scratch becomes infeasible.
>
> **Minor concerns:** We thank the Reviewer for their suggestions. We reformulated the expressions to avoid using the Saxon genitive and inverted the panel ordering in figures 3 and 4.
>
> [a] Satosi Watanabe, Information theoretical analysis of multivariate correlation. IBM Journal of Research and Development, 1960
>
> [b] Zihui Xue, Zhengqi Gao, Sucheng Ren, and Hang Zhao, The modality focusing hypothesis: Towards understanding crossmodal knowledge distillation. ICLR, 2023
>
> [c] Hamidreza Kamkari, Brendan Leigh Ross, Rasa Hosseinzadeh, Jesse C Cresswell, and Gabriel Loaiza-Ganem, A Geometric View of Data Complexity: Efficient Local Intrinsic Dimension Estimation with Diffusion Models. NeurIPS, 2024

---

> > ### Comment · Reviewer_i5Pr · 2024-11-30
> > **Thanks for reply**
> >
> > I would like to thank the authors for their clarifications, I look forward to read the paper with the extension to more modalities.
> >
> > Good luck!

---

### Official Review · Reviewer_ixgw · 2024-10-29

**Soundness:** 3
**Presentation:** 3
**Contribution:** 3
**Rating:** 8
**Confidence:** 3

**Summary:**

The authors introduce a novel metric for determining the correlation of high dimensional datasets. They propose to use normalized mutual information, whereby the mutual information, expressed in the form of entropies, is replaced with the intrinsic dimension, which they calculate using the Two-NN method. This metric is then tested on synthetic data against CCA and dCor, showing that IdCor can effectively find correlation in non-linear settings, and can do so better than dCor. The authors then go on to show this in the context of large vision models, showing the correlation between various large vision models, and then extend into a multi-modality setting, showing a correlation between vision models and LLMs on a multi-modality dataset.

**Strengths:**

From an originality/novelty perspective, the paper is quite novel in that this is a new use case for the relationship between intrinsic dimension and entropy, being substituted into a mutual information context.

The paper is quite clear, and it is very easy to see not only how various parts are put together, but the rationale between combining them as well. This is especially apparent in the beginning of Part 3: "Correlation through Intrinsic Dimension", where the authors give a very clear example why ID is needed, and the relationship betwen ID and NMI.

The authors do a very good job anticipating many of the questions readers might have, and explaining abnormalities in the results. In the results, they address several oddities that appear, such as the apparent lack of comparative performance on SigLIP and CLIP for coarsely aligned imagenet.

On the topic of results, the results are very good, with IdCor demonstrating very good performance, and a clear picture is painted as to its ability to deal with nonlinearity when obtaining a correlation metric.

The authors have a very substantial list of references, and are hitting many of the related and necessary works for understanding what the authors are trying to do.

The combination of these come together to create a very high-quality paper. Moreover, I believe this could have a moderate to highly significant impact on the field, and I can see how this new metric can certainly help with discovering the underlying mechanics of deep neural networks.

**Weaknesses:**

While the work is generally very clear, there is one place where a slight modification would help significantly.

While the reasoning behind using IdCor and not Id by itself is made fairly clear, the inclusion of the intrinsic dimension itself in table 1 could cause some confusion in this matter. A quick explanation of why Id is not sufficient (I believe no more than a sentence or two would be needed) should be all that is needed to ensure that it is clear that IdCor gives a better estimate, and is more useful.

Other than that, there are only a few very minor grammatical mistakes. The most pressing is Section I, last paragraph, bullet point 1,

**Questions:**

In table 3/Section 4.2.1 "Coarse Alignment", it is seen that IdCor is outperformed by other methods for CLIP and SigLIP. The authors do give an explanation as to why, stating that these two use a self-supervised loss, which have no explicit notion of class. This is a good explanation, but I think this should be discussed a little bit further. Is it that any model with no explicit notion of class/using a self-supervised loss will cause IdCor to be less performant (meaning this should be put as a limitation), or is this a fluke of these two specific examples, and on other methods with self-supervised loss, IdCor outperforms other metrics? Either a theoretical explanation of why this is the case, or a few more examples using self-supervised loss on coarsely aligned data could help greatly towards ensuring that future works do not encounter problems with IdCor by inadvertantly using it incorrectly.

---

> ### Author Response · Authors · 2024-11-19
>
> We are grateful to the Reviewer for their thoughtful feedback and for appreciating our work. We try to respond below to the weaknesses and questions that were presented in the review.
>
> **On the difference between $I_d$Cor and $I_d$:** We thank the Reviewer for suggesting to clarify this important point:  while the difference between the embedding dimension and the intrinsic dimension (alone) is a signature of the importance of the correlations within a dataset, it cannot be used for identifying correlations between different datasets, since the intrinsic dimension of the merged dataset would be affected both by the correlations within each dataset as well as those between the datasets. We now include this explanation in the introduction of our revised manuscript.
>
> **On the first bullet point:** We reformulated this bullet point in the revised version of our paper.
>
> **On coarse alignment in self-supervised models:** We thank the Reviewer for raising this point, as we were also intrigued by these results. We ran additional tests on another self-supervised vision-language model (VLM), BLIP [a], which has a similar architecture to CLIP but differs slightly in pre-training objective and data. As we report in our revised PDF, on this model $I_d$Cor produces a similar score to the one found in CLIP and SigLIP, even though in this case it is not surpassed by other metrics such as CKA and dCor. This consistency suggests that the pre-training objective plays a role in shaping the outcomes of our correlation metrics. This aligns with recent findings [b], which report significantly different representation similarity patterns in supervised vision models and VLMs. However, we do not see this as a limitation of our method, as correlation is still found with high confidence, as reported by the $p$-values, consistently at 0.01.
>
> [a] Junnan Li, Dongxu Li, Caiming Xiong, and Steven Hoi, Blip: Bootstrapping language-image pre-training for unified vision-language understanding and generation. ICML, 2022
>
> [b] Laure Ciernik, Lorenz Linhardt, Marco Morik, Jonas Dippel, Simon Kornblith, and Lukas Muttenthaler, Training objective drives the consistency of representational similarity across datasets. arXiv:2411.05561, 2024

---

> > ### Comment · Reviewer_ixgw · 2024-11-25
> >
> > Thank you for clarifications. I keep my original vote.

---

### Official Review · Reviewer_KtPo · 2024-10-29

**Soundness:** 2
**Presentation:** 3
**Contribution:** 3
**Rating:** 6
**Confidence:** 4

**Summary:**

This paper propose a way to define a proxy metric for mutual information to measure the relationship between two high dimensional distribution. The idea is fairly simple. Given distribution X and Y, and joint distribution X \times Y, it uses manifold learning based dimension reduction algorithm to reduce the dimension of X, Y and X \times Y. Intuitively, if X \times Y can be reduced to the same dimension of X or Y, then X and Y must be highly related. Moreover, the relationship between X and Y can be nonlinear due to the nonlinear dimension reduction algorithm.

They author shows clean and well illustrated toy example to illustrate how the metric identify the nonlinear relationship between data. And show the model can be applied to analyze the distance between two neural representations and multi-model neural representations.

**Strengths:**

1. Simple and well-motivated idea. I really like this idea!
2. Writing is good and easy to read.
3. Potentially influential application for deep learning and many field like neural science (distance between artificial/biological neural representation).

**Weaknesses:**

1. I think you want to show the discriminative power of I_dCor. This is probably the most important weakness. i.e. show I_dCor is not just a high score in general. If it maps different neural representation trained on imagenet to be close, then it should also map neural representations pre-trained on different dataset to be far away. So it would nice if you can come up with a task or metric that measure the discriminative power of your method I_dCor compared to other method. For example, construct subset of Imagenet using class label, map through a pretrained encoder, and use I_dCor to measure distance between the subset. You can use the distance metric and maybe just nearest neighbor to classify the label of each subset. Then show the I_dCor is superior then other measure. This might not be the best idea, but I believe you guys can come up with something better.

2. The assumption might be too strong. Like the author mentions, this idea is depends on the manifold learning algorithm for non-linear dimension reduction. The manifold learning must be able to identify the manifold from a high dimensional ambient space. But personally I think it's fine if you show the method actually works well, but you need more analyze to define "work well" than what you currently have. For example, see 1.

3. The analysis in figure 3 could be less ambiguous. I think you want to say the different (neural) representation of ImageNet is consistently similar under I_dCor, i.e. the similarity score has small variance. But it has a much bigger variance under a different metric under other metrics. I think this could be better shown than using two tables.

4. I'm not sure if figure 2 is that meaningful. Because these different metric are under different scale right? As the network becomes more nonlinear, other metric decrease, so is I_dCor. So I'm not sure what it really says.

**Questions:**

I don't have any questions.

---

> ### Author Response · Authors · 2024-11-19
>
> We thank the Reviewer for carefully evaluating our work and for appreciating the idea behind $I_d$Cor. We respond below to the weak points identified in the review.
>
> **W1 and W2:** We provide two additional experiments that we hope can clarify the discriminative power of $I_d$Cor, whose results are reported in the revised version of our manuscript in figures 12 and 13. In summary, in the first experiment we assess the correlation between image representations and multiple aligned textual representations, and show that $I_d$Cor successfully captures the decreasing strength of correlation as we progressively disrupt alignment between the two modalities. In the second experiment, we evaluate the correlation between visual representations computed on different image datasets, showing that in this case $I_d$Cor scores drastically decrease, while $p$-values increase. Regarding the experiment proposed by the Reviewer, from our understanding, we do not anticipate it being successful. The reason is that if the comparison involves distinct sets of image representations (even those with the same label), the $I_d$Cor score might reflect similarity but not true correlation. This is because there would be no correct alignment, even approximate, between the images, leading to high $p$-values and an unreliable score. However, if we have misunderstood any aspect of the proposed experiment, we are happy to participate in further discussions.
>
> **W3:** We agree that baseline methods exhibit higher variance between models compared to $I_d$Cor, and we added a sentence that mentions this in the main text. However, the main purpose of figure 3 is to provide a first large-scale evaluation (both in terms of data and variety of encoders) for $I_d$Cor, in a relatively controlled and well-known unimodal setting, in which baseline methods are known to perform reasonably well.
>
> **W4:** The Reviewer is right in saying that different metrics are under different scales. This is particularly evident as CKA and Distance Correlation produce scores well below 1 even in the linear case, due to their lack of invariance to invertible linear transformations ([a],[b]). However, the crucial point of this experiment is to show that, as the nonlinearity of correlation becomes more marked, baseline methods lose more signal, even in relative terms. In fact, while on average $I_d$Cor passes from 0.96 to 0.73 (losing 24% of the initial score), the closest performing baseline method, RBF kernel CKA, passes from 0.81 to 0.46, which represents a decrease of 43%.
>
>
> [a] Simon Kornblith, Mohammad Norouzi, Honglak Lee, and Geoffrey Hinton. Similarity of neural network representations revisited. ICML, 2019
>
> [b] Max Klabunde, Tobias Schumacher, Markus Strohmaier, and Florian Lemmerich. Similarity of neural network models: A survey of functional and representational measures. arXiv:2305.06329, 2023

---

> ### Comment · Reviewer_KtPo · 2024-11-25
>
> Thanks. The author addressed my concern and provide additional experiment to support its claim. I would like to raise my score. And recommending accept this paper.

---

### Official Review · Reviewer_2zy4 · 2024-11-03

**Soundness:** 3
**Presentation:** 2
**Contribution:** 2
**Rating:** 5
**Confidence:** 4

**Summary:**

The paper presents Intrinsic Dimension Correlation ($I_{d}$Cor), a novel similarity metric aimed at detecting correlations between high-dimensional data representations, particularly in nonlinear, multimodal contexts. Unlike traditional similarity metrics that rely on pairwise distances or explicit mappings, $I_{d}$Cor leverages intrinsic dimensionality (ID) as a proxy to assess correlation. It operates by measuring the reduction in combined intrinsic dimensionality when two datasets are concatenated, capturing their shared structural complexity. The paper applies $I_{d}$Cor to multimodal datasets, showing potential in revealing non-linear correlations between different modalities of data corresponding to the same source. $I_{d}$Cor demonstrates the ability to detect correlations that traditional metrics like CKA, CCA, and Distance Correlation (dCorr) might miss.

**Strengths:**

1. Innovative Use of Intrinsic Dimensionality for Correlation Measurement: Extending traditional distance correlation by looking at intrinsic dimension correlation is a novel perspective in representation similarity metrics.
2. Focus on Multimodal and Nonlinear Correlation: $I_{d}$Cor is specifically designed to tackle non-linear correlations in multimodal data, an area where traditional similarity metrics often struggle.
3. $I_{d}$Cor incorporates permutation testing to establish the statistical significance of its correlation scores. This approach adds robustness to the metric, allowing users to differentiate between meaningful correlations and spurious ones with a quantifiable p-value.

**Weaknesses:**

I believe the current version of the manuscript requires several improvements, but I’ve summarized my primary concerns below:

1. **Insufficient Evaluation:** $I_{d}$Cor’s evaluation lacks rigor in assessing fundamental metric properties. I would like to see a comprehensive study on the metric’s invariance properties (e.g., scaling, rotation, linear/non-linear transformations, translations, subset translations, etc.). Additionally, Ding et al. [1] proposed a comprehensive set of sensitivity and specificity tests for validating new similarity metrics. Performing these tests on $I_{d}$Cor would help establish its standing among existing metrics.


2. **Lack of Comparison to Recent, Relevant Metrics:**  $I_{d}$Cor is compared only to CCA and CKA, yet several newer similarity metrics have since emerged, such as Representation Topology Divergence (RTD) [2], Graph-Based Similarity (GBS)[3], and Inner Product Similarity[4]. You could even look at Deep CCA [5]. These metrics are known to capture complex structural and topological differences, often outperforming CKA and CCA. Benchmarking $I_{d}$Cor against these advanced metrics would clarify its effectiveness and limitations, especially in high-dimensional and multimodal contexts.

3. **Ambiguity in Purpose:** The paper’s positioning of $I_{d}$Cor as either a structural or functional similarity metric is unclear. It is referred to as a correlation metric, but intrinsic dimension calculation relies on distances between data points, suggesting a structural basis. However, the metric is evaluated on its ability to detect similarities between multimodal representations (which, according to Maiorca et al. [6], differ primarily by affine transformations), leaning toward functional similarity. The choice of benchmarking $I_{d}$Cor against structural similarity metrics like CKA (known to be limited to scaling and orthogonal invariances) instead of functional metrics contributes to this ambiguity. I recommend clarifying the intended classification in the paper.

4. **Limitations of TwoNN:**
- $I_{d}$Cor ’s reliance on intrinsic dimension estimation through second-nearest neighbor (TwoNN) consistency severely limits its insight into global structural alignment. I find it hard to see how $I_{d}$Cor could accurately quantify correlation between two spaces as a global value when it relies only on 2-NN consistency to assess similarity. Many fundamental changes can be made to a representation space while maintaining 2-NN consistency. Since the metric is “automatically invariant to any transformation that preserves the local neighborhood structure of data points,” it would miss changes that impact global structure but not local neighborhoods. This narrow focus makes $I_{d}$Cor unreliable in scenarios requiring a functional or comprehensive structural alignment.
- The paper evaluates $I_{d}$Cor on multimodal alignment with spaces that are already trained and tend to have good nearest neighbor consistency. I would like to see tests on spaces with high local but low global consistency to see if  $I_{d}$Cor can still perform well. This limitation raises the concern that  $I_{d}$Cor may be reporting high correlations due to a lack of sensitivity to broader transformations rather than genuinely revealing correlations among multimodal spaces. Additional experiments with controlled transformations (such as subset translations) are necessary to determine whether  $I_{d}$Cor’s high correlation scores reflect meaningful similarity or an inability to detect fundamental changes in the representation space.







[1] Frances Ding, Jean-Stanislas Denain, and Jacob Steinhardt. Grounding representation similarity
through statistical testing. Advances in Neural Information Processing Systems, 2021.

[2] Serguei Barannikov, Ilya Trofimov, Nikita Balabin, and Evgeny Burnaev. Representation topology
divergence: A method for comparing neural network representations. Proceedings of
the 39th International Conference on Machine Learning, 2022

[3] Zuohui Chen, Yao Lu, Jinxuan Hu, Wen Yang, Qi Xuan, Zhen Wang and Xiaoniu Yang. Graph-Based Similarity of Neural Network Representations. arXiv 2021

[4] Wei Chen, Zichen Miao, and Qiang Qiu. Inner product-based neural network similarity. In
Thirty-seventh Conference on Neural Information Processing Systems, 2023.

[5] Galen Andrew, Raman Arora, Jeff Bilmes, Karen Livescu.  Deep Canonical Correlation Analysis.  Proceedings of the 30th International Conference on Machine Learning, PMLR 28(3):1247-1255, 2013.

[6] Valentino Maiorca, Luca Moschella, Antonio Norelli, Marco Fumero, Francesco Locatello, and
Emanuele Rodola. Latent space translation via semantic alignment. Thirty-seventh Conference
on Neural Information Processing Systems, 2023.

**Questions:**

1. Have you run experiments where you examine more than two nearest neighbors of each point when computing intrinsic dimension?
2. What is the exact computational complexity of your proposed approach?
3. Why is only the p-value reported for coarse vs. random alignment? I would like to see the IdCor value as well.
4. Are the p-values consistently the lowest possible in all the multimodal heatmaps?
5. For the N24News dataset, do you treat the headline as the corresponding text component or the caption?

---

> ### Author Response · Authors · 2024-11-19
>
> We thank the Reviewer for their feedback on our work, and respond here to the weaknesses and questions outlined in the review.
>
> **W1:** The invariances of the $I_d$Cor method strongly depend on the intrinsic dimension estimator employed. When using the TwoNN estimator, the method is inherently invariant to rotation, translation, and global scaling. Additionally, due to its connection with mutual information, we expect $I_d$Cor to exhibit a significant degree of invariance to nonlinear transformations. This expectation is supported by Figure 2, which tests invariance as a function of the nonlinearity of the transformation. Although we have not specifically tested subset translation, it is likely that, as far as the points of the subset are clustered in the same region, this should not impact the TwoNN estimate, which relies on distances between the two nearest neighbors.
>
> Regarding the tests proposed by Ding et al. [a], we do not anticipate meaningful results when applying them to $I_d$Cor. These tests are designed to evaluate indices that measure geometric similarities between representations, whereas our method quantifies the amount of shared information among representations. This fundamental difference affects the applicability and relevance of such tests. For instance, the sensitivity test would not be effective, as all principal components are linear combinations of the original coordinates and therefore retain their information by definition.
>
> As noted in the last paragraph of section 4.2, our method focuses on detecting statistical dependencies between different representations rather than measuring geometric similarities. This does not imply that the index always yields a high value; rather, when measuring similarities between transformations, it is only sensitive to those that disrupt the flow of information, such as random shuffling of pairings.
>
> In the revised version, we propose a new test to further illustrate this point. Specifically, we use a dataset of images with five possible captions to investigate whether the representations of this multimodal dataset are informative about one another. We then disrupt the correlations between captions and images by progressively shuffling the pairings and observe the following (see figure 12 in the revised manuscript): 1) a high correlation score when captions are correctly paired, 2) a gradual decrease in correlation as the pairings are partially shuffled, and 3) a low correlation score with high p-values, indicating low significance, when pairings are completely randomized.
>
> It is worth noting that in these experiments, the $I_d$ of the merged dataset ranges from 40 to 80. This suggests that our method may encounter limitations due to the constraints of the intrinsic dimension estimator. Nonetheless, the observed trends align well with our theoretical expectations.
>
> **W2:** We thank the reviewer for pointing us to these works. We computed the divergence scores using the Representation Topology Divergence (RTD) method [b] for the multimodal N24News representations we consider in figure 4 of our paper. Results showcase a clear disparity between modalities: excluding the diagonal (comparison of a model with itself), the average divergence among image encoders is 26.8, while the average scores for text encoders and across modalities are respectively 45.2 and 45. These results suggest that RTD is capturing a much stronger similarity in the visual modality than in the textual and multimodal settings. We did not include the full results in the current version of the manuscript because they are not directly comparable with $I_d$Cor, nor with the other baselines. In fact, RTD produces an unbounded divergence score, and not a correlation score, and we did not find any obvious way of scaling the results in a comparable range (simple min-max scaling would not be fair as it would imply zeroing-out a non null result). However, we are open to reconsider this choice in the final version of the paper if the Reviewer deems it appropriate.
>
> **W3:** We agree with the Reviewer that the positioning of $I_d$Cor in this respect should be made more clear. We take the findings of [c] as a starting point for our investigation. In fact, while they observe that multimodal representations can be functionally similar (affine transformations are sufficient to map one modality into the other with satisfactory performance on a downstream task), which should imply some degree of correlation between those, we notice that current correlation (structural) methods widely employed to analyze neural representations capture weak to no correlation in the multimodal setting. This motivates us to propose an information-based correlation method to overcome the limitations of existing methods. We refactored the introduction of our paper to clarify this point.

---

> > ### Author Response · Authors · 2024-11-19
> >
> > **W4:**
> >
> > - There are several reasons for using TwoNN as $I_d$ estimator. Among them, let us highlight four: 1) It is thought for nonlinear manifolds, as the ones expected in representations of deep models; 2) it is simple and fast to compute, allowing to obtain the $p$-value through permutation testing in reasonable times; 3) there is no need to tune any hyperparameter and 4) it is widely used in the analysis of neural representations (e.g., see references [d], [e], [f]). However, $I_d$Cor is independent from the $I_d$ estimator and could, in principle, be applied using any other estimator. We prove this point by running experiments with the Maximum Likelihood Estimator [g], setting the number of neighbors to 100 (that we judge as big enough to surpass the pure local estimator) obtaining similar or slightly worse results compared to the ones obtained when using TwoNN (results are provided in section A.3.1 of the revised PDF).
> >
> > - We were not able to identify a dataset with the suggested characteristics. If the Reviewer can provide a proposal we will be happy to test it. However, we anticipate the Reviewer being right in hypothesizing the lack of sensitivity of our method to these global transformations, unless they disrupt the information flow. The key point is that we are not providing a structural similarity score, but a measure of the information shared among different representations. Therefore, we would not consider invariance to global transformations as a weakness (as far as there is a way to trace the information flow between the original and the transformed representations), but a demonstration that the method can trace the information transfer even when applying these highly nonlinear transformations.
> >
> > **Q1:** Yes, we ran experiments using a different estimator (Maximum Likelihood Estimator-MLE [g]), which allows tuning the number of nearest neighbors. Results are reported in section A.3.1 of the revised manuscript.
> >
> > **Q2:** The complexity of our algorithm depends crucially on that of the intrinsic dimension estimator of choice, as $I_d$Cor requires S+3 calls to the estimator (3 to obtain the coefficient, S to obtain the associated $p$-value), where S is the number of shuffles to perform for the permutation test. In the case of the TwoNN estimator, whose complexity is $O(NlogN)$ [h], the complexity of $I_d$Cor becomes $O(SNlogN)$.
> >
> > **Q3:** We thank the Reviewer for the suggestion and report the $I_d$Cor scores in the revised manuscript.
> >
> > **Q4:** Yes, as we state in the manuscript at the beginning of the Results section, when $p$-values are not explicitly reported, they are always the lowest possible ($\frac{1}{101}$, since we are considering 100 permutations for $p$-value computation).
> >
> > **Q5:** We consider the captions as the text components of N24News.
> >
> > [a] Frances Ding, Jean-Stanislas Denain, and Jacob Steinhardt, Grounding representation similarity through statistical testing. NeurIPS, 2021
> >
> > [b] Serguei Barannikov, Ilya Trofimov, Nikita Balabin, and Evgeny Burnaev, Representation topology divergence: A method for comparing neural network representations. ICML, 2022
> >
> > [c] Valentino Maiorca, Luca Moschella, Antonio Norelli, Marco Fumero, Francesco Locatello, and Emanuele Rodolà, Latent space translation via semantic alignment. NeurIPS, 2023
> >
> > [d] Alessio Ansuini, Alessandro Laio, Jakob H Macke, and Davide Zoccolan, Intrinsic dimension of data representations in deep neural networks. NeurIPS, 2019
> >
> > [e] Lucrezia Valeriani, Diego Doimo, Francesca Cuturello, Alessandro Laio, Alessio Ansuini, and Alberto Cazzaniga, The geometry of hidden representations of large transformer models. NeurIPS, 2023
> >
> > [f] Ilya Kaufman and Omri Azencot, Data representations’ study of latent image manifolds. ICML, 2023
> >
> > [g] Elizaveta Levina and Peter Bickel. Maximum likelihood estimation of intrinsic dimension. NeurIPS, 2004
> >
> > [h] Elena Facco, Maria d’Errico, Alex Rodriguez, and Alessandro Laio, Estimating the intrinsic dimension of datasets by a minimal neighborhood information. Scientific Reports, 2017

---

> > > ### Comment · Reviewer_2zy4 · 2024-11-26
> > >
> > > Thank you for your extensive rebuttal. I appreciate the time and effort you’ve put into addressing the points raised during the review process.
> > >
> > > While I am largely satisfied with the responses presented, I remain unconvinced by the choice of the TwoNN estimator for IdCor. However, since IdCor is flexible enough to support other intrinsic dimension estimators, I believe it remains a valuable measure.
> > >
> > > I would suggest exploring the following synthetic dataset for further validation. For example, consider a dataset where points are clustered in groups of three (perhaps on the surface of a sphere), and these clusters are shuffled around. Such a scenario would introduce significant changes in functional similarity and information flow, particularly for tasks that depend on contextual representations. It would be interesting to see whether IdCor with TwoNN estimator effectively reflects these changes.
> > >
> > > I would also encourage the authors to review the work by David J.C. MacKay and Zoubin Ghahramani, "Comments on 'Maximum Likelihood Estimation of Intrinsic Dimension' (2005)", which proposes a superior intrinsic dimension estimator compared to the maximum likelihood estimator. It might provide additional insights or improvements.
> > >
> > > Thank you for reproducing the experiment using RTD. Since RTD is gaining popularity, it might be worth including the results in the appendix for readers who may have similar questions or concerns in the future.
> > >
> > > In light of the responses presented by the authors, I am raising my score to 5.

---

> ### Author Response · Authors · 2024-11-27
>
> We thank the Reviewer again for their valuable suggestions and proposed improvements to our work.
>
> We evaluated $I_d$Cor in the suggested synthetic setting by generating a dataset of 3000 points organized into clusters of three. To create the dataset, we first sampled 1000 "anchor" points uniformly from the surface of the unit sphere. Each anchor point was then perturbed with three independent samples of Gaussian noise (mean $0$, standard deviation $0.01$) to form its associated cluster. Finally, we used $I_d$Cor to measure the correlation between this dataset and a modified version in which the clusters were randomly shuffled. $I_d$Cor with the TwoNN estimator is able to capture a decrease in correlation with respect to the non-shuffled case, reporting a coefficient of $0.62$ ($p$-value $0.01$). As one could expect in this setting, when using a "less-local" intrinsic dimension estimator, such as MLE with  $k=100$ (number of neighbors), $I_d$Cor detects a more pronounced drop in correlation, yielding a coefficient of $0.29$ ($p$-value $0.01$).
>
> We thank the Reviewer for pointing us to the work by MacKay and Ghahramani on the Maximum Likelihood Estimator. We will consider this improved estimator for future developments of our work.
>
> Following the Reviewer’s suggestion, the new revision of our manuscript contains the full results of RTD on multimodal N24News representations in the Appendix.

---

### Official Review · Reviewer_xZgg · 2024-11-07

**Soundness:** 3
**Presentation:** 3
**Contribution:** 3
**Rating:** 8
**Confidence:** 4

**Summary:**

The paper introduces a new metric, Intrinsic Dimension Correlation (IdCor), designed to detect nonlinear correlations between features describing data points, in particular the latent representations output by deep networks. IdCor leverages "intrinsic dimension" (Id) as a proxy for information and estimates correlation using a normalized mutual information approach. By quantifying correlation based on the minimal number of variables required to represent data, IdCor aims to provide insight into nonlinear dependencies that standard methods may not achieve. The authors test their method on a variety of synthetic datasets and deep neural networks, including multimodal models.

**Strengths:**

- IdCor introduces an interesting approach of measuring nonlinear correlations by combining the notions of mutual information and intrinsic dimensionality.

-  The paper conducts experiments using both synthetic and real-world datasets.

- The paper is well-written and organized.

- Interesting discussion on latent space alignment.

**Weaknesses:**

__Choice of Id estimator:__ Not enough reasons are given for the choice of using TwoNN. Given that this is the main point of the paper, benchmarking it against other more popular methods for Id estimation would be strongly advised, e.g., Levina & Bickel's. Moreover, through lines 179--185, the authors list several references that estimated the Id found in the representations output by deep networks. I believe it would be helpful to list which methods those authors used for computing Id and possibly using the same ones to compare against using TwoNN.

__Id assumptions:__ The proposed measure IdCor assumes that the intrinsic dimensionality is roughly constant throughout the data. That is unlikely to be true for most real-world data (in line with what the authors point out in lines 312--313). A simple example is one in which we have clustered data but different clusters might have different dimensionalities -- a reasonable assumption if one takes into account that a clustered representation might be clustered precisely due to the fact that different "concepts" may require different topologies. E.g., the dimensionality of the space of images of dogs is likely to much much higher than that of images of soccer balls due to the different symmetries and "directions" of variability (e.g. color) between the two objects encountered in natural scene data. Here are a couple of recent papers that directly consider this possibility and propose solutions for dealing with this dimensional heterogeneity:

[1] Allegra, M., Facco, E., Denti, F., Laio, A., & Mira, A. (2020). "Data segmentation based on the local intrinsic dimension". Scientific reports, 10(1), 16449.

[2] Dyballa, L., Zucker, S., (2023). “IAN: Iterated Adaptive Neighborhoods for manifold learning and dimensionality estimation”, Neural Computation, 35 (3): 453-524.

The first one is in fact by the same group as that of the TwoNN method. So I would consider both including this limitation in Discussion section, and proposing a way to adapt the IdCor computation that would allow for multiple dimensionalities to be present within the same dataset. (E.g., considering each subset with homogeneous dimension at a time and later aggregating the results somehow seems like a reasonable candidate strategy.)

__Synthetic results:__ I believe the authors jump prematurely into the deep net section without sufficiently exploring the synthetic data first (important, because it provides ground truth). For example: since the data is randomly sampled, how many times has this experiment been repeated. Then, values reported in Table 2 should show the mean and standard. dev. across multiple random seeds, not just one instance. (as far as I understand it, although multiple permutations were considered to compute the p-values, they all came from the same data set, correct?)  How many points are sampled? (Important when estimating dimensionality.) How robust are the results w.r.t. the average sampling density? How is the sampling done? Uniformly or drawn from a Gaussian distribution? (both should be tested) Why not show the results for several values of Id? I think the results with synthetic data can be made considerable more interesting but adding these. These modifications are easy to implement (see [2] for examples), and it would be a much stronger way of empirically testing the robustness of the proposed method. (I would also suggest to move these results into the Results section.)

__Figure 2:__ the points plotted should also be the mean across several different weight initializations of the MLP, with std. dev. shown as error bars.

__Figure 3:__ Authors show that the value of IdCor are high when the same dataset is compared across multiple representations output from networks. I believe it would be important to show that if the dataset is changed, then IdCor should go down considerable (this would work as a sanity check that the method is not simply giving a high correlation between any two high-dimensional datasets). The multimodal results in Table 3 do not help in showing this, since the values produced by IdCor are still the highest (or close to) every case.

__Minor points:__

Line 239-240: It is confusing to call these datasets "2D", given that two of them are actually 1-D manifolds. What is common between them is that they are all embedded in 2D space;  this should be made more clear. Line 196: for the same reason, it is inadvisable to call the spiral shaped dataset simply a "2D dataset". It is a 1-D manifold embedded in 2-D ambient space. This is in fact what was shown in Fig. 1, bottom-right: the Id of the circle is correctly labeled as 1.

The paper could elaborate more on how IdCor’s detection of nonlinear correlations leads to interpretability benefits, providing examples of how this information could aid in model evaluation or adjustment. In other words, clarifying how quantifying similarity between different models (lines 36--37, 107)) can help interpreting them?

TwoNN should be defined either in the main text or the Appendix since it is the method underlying all Id computations used in IdCor.

Lines 163-164: PCA might not be correct even if the data lie in a hyperplane: for example, a circle (1-dimensional) would lie on a hyperplane yet would require 2 principal component coordinates to be embedded. Thus, PCA can find the minimum number of dimensions required to embed the data (dimension of the ambient space), but not necessarily their intrinsic dimension (in the topological sense).

The methods listed in lines 166--169 can be used for dimensionality reduction but not for dimensionality estimation. Unlike PCA, which produces a way to rank dimensions based on their explained variance, all of these require the user to pre-specify the desired dimension.

**Questions:**

- How does IdCor perform under different levels of added noise, especially when applied to real-world, noisy datasets?

- Why is Id+ not included in Table 2?

---

> ### Author Response · Authors · 2024-11-19
>
> We thank the Reviewer for their insightful feedback on our work and for suggesting several potential improvements to it. We try to address here the main weaknesses and questions that emerged in the review.
>
> **Choice of $I_d$ estimator:** We agree that this is a crucial point, as our method fundamentally relies on $I_d$ estimation. We did experiment with Levina and Bickel’s MLE estimator [a], which is in fact implemented in the attached code in `utils/intrinsic_dimension.py`, but did not find outstanding differences with respect to TwoNN (results are similar or slightly less accurate in the controlled synthetic settings). Moreover, MLE has the disadvantage that it requires careful tuning of a hyperparameter, the number of nearest neighbors to consider. However, we provide results using MLE in the same synthetic settings of tables 1 and 2 in the Appendix (section A.3.1) of our revised PDF. Regarding the $I_d$ estimators employed by previous works on neural representations, references [b-e] only used TwoNN, [f] used both TwoNN and MLE, while [g] used a dozen of estimators, including TwoNN and MLE.
>
> **$I_d$ assumptions:** We agree that the assumption of a uniform intrinsic dimension value over the entire dataset may not hold universally across all types of datasets. A potential solution, which requires the ability to reliably estimate $I_d$ locally, is to calculate the correlation score locally as well. This would allow for the computation of an $I_d$Cor score at the cluster level (by using Hidalgo [h]) or even at the individual point level (using IAN [i]). We have now clarified this point in the discussion section and included a proof-of-concept illustration of local $I_d$Cor in the Appendix. Moreover, we would like to thank the Reviewer for presenting to us the IAN method, which we were not aware of.
>
> **Synthetic results:** We thank the Reviewer for raising these points. Yes, even though $p$-value computation involves repeated estimations of $I_d$ on permuted data, this does not involve resampling data. We now provide a more detailed evaluation in the proposed synthetic settings: all results are reported in terms of mean$\pm$standard deviation over multiple independent random samplings of data (with the exception of $p$-values, that are expressed in aggregate form as $[min, max]$ ranges) and we provide in the Appendix results with a different estimator (in section A.3.1, see the first weakness), with a different data sampling distribution (A.3.2), and when noise of increasing magnitude is applied to data (A.3.3). Moreover, we moved the main text section on synthetic experiments in the Results section, following the Reviewer’s suggestion.
>
> **Figure 2:** We updated this figure in the revised version of our manuscript to include results from 10 independent random initializations of the MLP.
>
> **Figure 3:** We followed this suggestion and added an experiment on correlating visual representations produced by the same model (CLIP-ViT-B) on different input datasets. The results are reported in figure 13 of the revised manuscript and they show that, as one would expect, $I_d$Cor score significantly drops when applied to different and uncorrelated representations, while the $p$-values increase. Moreover, in a similar spirit, we produced another experiment on multimodal Flickr30k representations, whose results are reported in figure 12. In this test, we pair each image representation with 5 caption representations, and show that $I_d$Cor consistently decreases as we progressively break the correlation between visual and textual representations.
>
> **Minor points:**
>
> - We adjusted the terminology to remove the source of confusion about 2D datasets.
> - We agree that it would be interesting to accommodate a more detailed explanation of the positive impact that representation similarity analysis can have on interpretability, which is now limited to the second paragraph of section 2.1. However, due to space constraints, we are unable to extend our review.
> - We completely agree with the Reviewer. The TwoNN method is now described in the Appendix.
> - We reformulated the sentence in an attempt to clarify the point on $I_d$ estimation with PCA.
> - We agree with the Reviewer and therefore we removed the references to dimensionality reduction methods.
>
> **Q1:** As anticipated above, we included an analysis of the robustness of $I_d$Cor to noise in the Appendix of our revised manuscript, in section A.3.3.
>
> **Q2:** Following the suggestion, we included $I_d\oplus$ in table 2.

---

> > ### Author Response · Authors · 2024-11-19
> >
> > [a] Elizaveta Levina and Peter Bickel. Maximum likelihood estimation of intrinsic dimension. NeurIPS, 2004
> >
> > [b] Alessio Ansuini, Alessandro Laio, Jakob H Macke, and Davide Zoccolan, Intrinsic dimension of data representations in deep neural networks. NeurIPS, 2019
> >
> > [c] Lucrezia Valeriani, Diego Doimo, Francesca Cuturello, Alessandro Laio, Alessio Ansuini, and Alberto Cazzaniga, The geometry of hidden representations of large transformer models. NeurIPS, 2023
> >
> > [d] Ilya Kaufman and Omri Azencot, Data representations’ study of latent image manifolds. ICML, 2023
> >
> > [e] Bradley CA Brown, Jordan Juravsky, Anthony L Caterini, and Gabriel Loaiza-Ganem, Relating regularization and generalization through the intrinsic dimension of activations. NeurIPS Workshop on Optimization for Machine Learning, 2022
> >
> > [f] Henry Kvinge, Davis Brown, and Charles Godfrey, Exploring the representation manifolds of stable diffusion through the lens of intrinsic dimension. ICLR Workshop on Mathematical and Empirical Understanding of Foundation Models, 2023
> >
> > [g] Emily Cheng, Corentin Kervadec, and Marco Baroni, Bridging information-theoretic and geometric compression in language models. EMNLP, 2023
> >
> > [h] Michele Allegra, Elena Facco, Francesco Denti, Alessandro Laio, and Antonietta Mira, Data segmentation based on the local intrinsic dimension. Scientific Reports, 2020
> >
> > [i] Luciano Dyballa and Steven W Zucker, Ian: Iterated adaptive neighborhoods for manifold learning and dimensionality estimation. Neural Computation, 2023

---

> > > ### Comment · Reviewer_xZgg · 2024-11-25
> > >
> > > I thank the authors for the clarifications. Most of my concerns have been addressed (as well as those from other reviewers) and I believe the manuscript has improved considerably, so I am raising my score to 8.

---

### Author Response · Authors · 2024-11-19

We are sincerely grateful to all Reviewers for their thoughtful feedback and suggestions, which we believe are very beneficial for our work.  We particularly appreciate that they found the idea of using intrinsic dimension to quantify correlation novel and effective and the manuscript clear and easy to follow.

We tried to address the weaknesses and questions raised by each Reviewer in the individual replies, and we provide here a comprehensive list of the revisions we applied to our manuscript:


- We revisited the introduction, clarifying the positioning of our proposed metric with respect to existing methods (`2zy4`) and the fact that intrinsic dimension ($I_d$) alone is not sufficient to compute correlations (`ixgw`);
- We reformulated and clarified the first bullet point in our list of contributions (`ixgw`);
- We made some corrections to the section on related works in intrinsic dimension estimation (`xZgg`);
- We moved results on synthetic data to the Results section, and provided additional results on multiple random samplings of data, using a different $I_d$ estimator, on different data distributions and in presence of noise. We also included $I_d\oplus$ in table 2 (`xZgg`);
- We updated figure 2 to include results from multiple random initializations of the MLP (`xZgg`);
- We included an additional self-supervised vision encoder (BLIP) in our experiments on ImageNet data (`ixgw`);
- We removed occurrences of Saxon genitive and inverted panel ordering in figures 3 and 4 (`i5Pr`);
- We modified table 3 to include results on multiple independent random shufflings of data and added $I_d$Cor coefficients (`xZgg`, `2zy4`);
- We included two new experiments: one on the correlation between representations obtained on different datasets and one on multimodal correlation between visual representations and partially aligned textual representations (`KtPo`, `xZgg`, `2zy4`);
- We clarified the paragraph on future directions by adding a reference to Total Correlation (`i5Pr`);
- We included a short description of the TwoNN estimator in the Appendix (`xZgg`);
- We provided a proof-of-concept experiment on local correlation in datasets with variable intrinsic dimensionalities (`xZgg`).

Revision (November 27):

- We added RTD results on multimodal representations in the Appendix (`2zy4`).

---

### Public Comment · ~James_Bailey1 · 2025-02-27
**Earlier literature on using intrinsic dimension to assess dependency**

I believe this paper may be unaware of some earlier literature that already developed a theoretical basis for using intrinsic dimensionality to assess dependency and which has proposed estimators.  The theory is built on the (alpha)-Renyi dimension. Please see:

1) Mutual information dimension (MID)
Sugiyama, M., Borgwardt, K. M.: Measuring Statistical Dependence via the Mutual Information Dimension, Proceedings of the 23rd International Joint Conference on Artificial Intelligence (IJCAI 2013),
This covers the bivariate case.

2)
Measuring Dependency via Intrinsic Dimensionality. Simone Romano, Oussama Chelly, Xuan Vinh Nguyen, James Bailey and Michael E. Houle. Proceedings of the 23rd International Conference on Pattern Recognition (ICPR), December 4-8, Cancun, Mexico, 2016.
This covers the multivariate+normalized case.

---

> ### Public Comment · ~Alex_Rodriguez1 · 2025-03-01
>
> Thank you very much for your comment. We were not aware of these papers and, therefore, we modified the last version of the paper accordingly .

---

### Meta-Review · Area_Chair_tUNx · 2024-12-16

**Metareview:**

This paper proposed a new definition of correlation between two data manifolds based on an estimator of intrinsic dimensionality, namely the IdCor, and tested it on synthetic and multimodal data and revealed better similarity between paired embedding across modalities. All reviews recognized the simple and reasonable definition. The paper has been improved substantially in the rebuttal phase.

The reviewers further discussed the limitation of this work, which provides a definition without discussing its basic properties with theoretical arguments. The reviewers suggested to discuss how perturbations affect IdCor and its invariance, and discuss whether a high value of IdCor actually identifies correlations, and support the correlation detection with downstream applications like transfer learning or model stitching. The authors are highly recommended to address these comments in the final version.

**Additional Comments On Reviewer Discussion:**

All reviewers have responded to the authors' rebuttal or participated in the discussion.

The overall rating is the highest in my batch. The only reviewer on the (weak) rejection side, Reviewer 2zy4, clearly explained their concerns: the lack of theoretical analysis and lack of downstream applications to support the definition, which is agreed by at least one other reviewer and myself.

However, these two reviewers and myself also acknowledged on the novel and interesting definition, which could spark discussion in the ICLR community.

Among the revisions that the authors have made, they have provided a comparison with more recent metrics (RTD) in the appendix to address the reviewers' comments.

---

### Decision · Program_Chairs · 2025-01-22

Accept (Poster)